# Bacteriophage therapy for the treatment of *Mycobacterium tuberculosis* infections in humanized mice
Fan Yang[1,2,3,8], Alireza Labani-Motlagh[1,2,3,7,8], Jose Alejandro Bohorquez [1,2,3,8], Josimar Dornelas Moreira[2,3], Danish Ansari[1,2,3], Sahil Patel[1,2,3], Fabrizio Spagnolo[4], Jon Florence[2,3], Abhinav Vankayalapati[2,3], Tsuyoshi Sakai[2,3], Osamu Sato[2,3], Mitsuo Ikebe[2,3], Ramakrishna Vankayalapati[2,3], John J. Dennehy [5,6] ✉, Buka Samten [2,3] ✉ & Guohua Yi [1,2,3] ✉

The continuing emergence of new strains of antibiotic-resistant bacteria has renewed interest in phage therapy; however, there has been limited progress in applying phage therapy to multi-drug resistant *Mycobacterium tuberculosis* (*Mtb*) infections. In this study, we show that bacteriophage strains D29 and DS6A can efficiently lyse *Mtb* H37Rv in 7H10 agar plates. However, only phage DS6A efficiently kills H37Rv in liquid culture and in *Mtb*-infected human primary macrophages. We further show in subsequent experiments that, after the humanized mice were infected with aerosolized H37Rv, then treated with DS6A intravenously, the DS6A treated mice showed increased body weight and improved pulmonary function relative to control mice. Furthermore, DS6A reduces *Mtb* load in mouse organs with greater efficacy in the spleen. These results demonstrate the feasibility of developing phage therapy as an effective therapeutic against *Mtb* infection.

*Mycobacterium tuberculosis* (*Mtb*), the causative pathogen of Tuberculosis (TB), infected 10.6 million people and caused 1.3 million deaths globally, making it one of the leading causes of death by a single infectious agent. More importantly, one-fourth (~1.7 billion) of the world's population has been latently infected with *Mtb* (LTBI)[1]. When complicated with other co-morbid conditions, such as diabetes, HIV, and COVID-19, the morbidity and mortality of tuberculosis infection is further increased[2]. Although these devastating facts highlight the threat posed by this deadly bacterium, the emergence of drug resistant *Mtb* strains in recent years has worsened the situation in terms of *Mtb* prevention and control.

Upon establishment of infection, *Mtb* exhibits remarkable abilities to adapt to the local environment to evade the host's immune responses due to its unique and dynamic four-layer cell envelope structure, which can help it adapt to hostile lung microenvironments and facilitate its entry into a non-replicating drug-tolerant persister state. In this state, the bacilli are well protected against antibiotic therapy[3,4], and are potentially a source of new strains of drug-resistant *Mtb* (DR-TB)[5]. In 2019, around half a million

people developed DR-TB, and 78% of these cases developed multidrug-resistant TB (MDR-TB; resistant to first-line drugs, rifampicin and isoniazid) or extensively drug-resistant TB (XDR-TB; resistant to rifampicin, isoniazid, and a second-line drug)[6]. MDR-TB and XDR-TB are more challenging to treat and are associated with increased morbidity and mortality[7]. Therefore, the development of alternative therapies in addition to antibiotics is critical for the advancement of TB therapy in the era of *Mtb* drug resistance.

Bacteriophage therapy has emerged as a renewed approach to eliminate bacterial infections[8,9]. Bacteriophages (also known as phages), viruses that infect bacteria, are bacteria's natural enemies and have been used to control bacterial infections since even before the discovery of antibiotics[10]. In the age of antibiotic resistance, phage therapy has drawn tremendous attention. Recent clinical cases and trials demonstrated that phages can be used to treat antibiotic-resistant bacterial infections with positive clinical outcomes[11–16], and patients with drug-resistant *M. abscessus* and *M. chelonae* have been successfully treated[17–19]. These prominent bodies of work on

[1]Department of Medicine, The University of Texas at Tyler School of Medicine, Tyler, TX, USA. [2]Center for Biomedical Research, The University of Texas Health Science Center at Tyler, Tyler, TX, USA. [3]Department of Cellular and Molecular Biology, The University of Texas Health Science Center at Tyler, Tyler, TX, USA. [4]Life Sciences Department, Long Island University Post, Brookville, NY, USA. [5]Biology Department, Queens College of The City University of New York, Flushing, NY, USA. [6]The Graduate Center of The City University of New York, New York, NY, USA. [7]Present address: Center for Discovery and Innovation, Hackensack Meridian Health, Hackensack, NJ, USA. [8]These authors contributed equally: Fan Yang, Alireza Labani-Motlagh, Jose Alejandro Bohorquez. ✉e-mail: John.Dennehy@qc.cuny.edu; Buka.Samten@uthct.edu; Guohua.Yi@uthct.edu

using phage therapy to treat other mycobacterial infections inspired us to pursue phage therapy as a potentially effective treatment for *Mtb* infection. However, as to the phage therapy for *Mtb* treatment, the significant advancements have focused on screening lytic phages for effective killing of *Mtb* in bacterial culture plates, demonstrated by plaque formation[20–22], but no systematic study has characterized the effectiveness of phages in killing *Mtb* in primary human macrophages or in reliable animal models. Therefore, extensive preclinical studies, especially using primary human macrophages or animal models that resemble human clinical settings, are of exceptional clinical importance in the development of phage therapy for *Mtb* treatment.

One of the major challenges to eliminating *Mtb* infection is *Mtb*'s induction of granuloma formation once host defenses fail to kill the bacteria. While granuloma formation limits *Mtb* growth, it provides a survival niche for *Mtb* replication when the immune system is weakened. Granulomas, the hallmark of TB pathology, consist of macrophages, neutrophils, and lymphoid cells, including T and B cells, and are formed upon *Mtb* infection in patients[23]. Most commonly employed mouse models can be infected with *Mtb*, but *Mtb* can only form a granuloma-like structure in mice. Mouse *Mtb* infections lack the caseous necrotic granulomas that are often observed in TB patients[24]. The Kramnik's mouse can form hypoxic, encapsulated granulomas with a caseous necrotic center following *Mtb* infection[25,26], but generally these mouse models cannot fully recapitulate the human immune responses. Recently, humanized mice based on NOD-scid IL2Rgamma[KO] (NSG) mice were developed. However, the innate immune systems in these mice were deficient and transplanted human hematopoietic stem cells (hHSCs) were generally not well-developed; thus, the human B, T, and myeloid cells were immature, and the NK cells lost functions[27,28]. These mouse models provide valuable tools to study *Mtb* infection[24,29]; however, the myeloid barrier in these NSG mice causes a relative lack of leukocyte differentiation[28]. The lack of sufficient granuloma formations (which requires mature macrophages) makes these mice less able to recapitulate *Mtb* infection in humans.

These deficiencies are rectified in the newly developed NSG-SGM3 mice, which can transgenically express three human cytokine/chemokine genes IL-3, GM-CSF, and KITLG. The expression of these genes can enhance the differentiation and maturation of the myeloid-lineage cells[28,30–33]. Moreover, these three transgenic genes can improve the homeostasis of human CD34+ HSCs, increase neutrophil and macrophage numbers and function, and stimulate hCD34+ cells to differentiate into myeloid progenitor cells[32,34]. Thus, this humanized mouse model can generate sufficient and fully functional myeloid cells, including macrophages, to study *Mtb* infection in the context of an entire human immune system.

In this study, we show that phage DS6A can efficiently kill wild-type *Mtb* H37Rv in vitro (in both 7H10 agar plates and 7H9 liquid culture), in primary human macrophages, and in humanized NSG-SGM3 mice, demonstrating the feasibility of developing phage DS6A as an effective therapeutic against *Mtb* infection.

## Results
### Phage infectivity in *M. smegmatis* and in *M. tuberculosis* H37Rv
We first tested three bacteriophages, D29, Chah and DS6A, for their lysis capacities of *M. smegmatis* and *Mtb* H37Rv by plaque assay (Chah was used as a negative control for *Mtb* killing[35]). Serially diluted phages ($10^2$–$10^4$ pfu) were incubated with *M. smegmatis* or H37Rv in 12-well plates with 7H10 agar. Infection was carried out at a multiplicity of infection (MOI, a ratio of phage to bacteria) between 0.001 and 0.1. The results showed that two phage strains, D29 and Chah lysed *M. smegmatis* (Fig. 1A) and formed different sized plaques, and two phage strains D29 and DS6A, lysed H37Rv and formed plaques (Fig. 1B).

We then tested the *Mtb* killing ability of these three phage strains in liquid culture by measuring the optical density at 600 nm ($OD_{600}$) of the H37Rv suspensions at different time points after incubation with phages at an MOI of one, by inoculation of 7H9 media with $5 \times 10^4$ CFUs of H37Rv and $5 \times 10^4$ pfu of phages. We found that only DS6A significantly reduced

*Mtb* $OD_{600}$ on day 9 (Fig. 1C), while other phages did not affect *Mtb* $OD_{600}$ on day 9, although D29 also showed the ability to kill *Mtb* in agar plates. We then plated the cultures from different time points to determine the CFU of the bacilli (Fig. 1D, E). The CFU results were consistent with the $OD_{600}$ measurements. To exclude the possibility that the unsuccessful *Mtb* elimination of D29 in liquid culture is due to the low infection titer, we used MOIs of 1, 10, and 100 to infect the *Mtb* in 7H9 culture, and monitored the *Mtb* CFU over time. We found that even the highest MOI (MOI of 100) did not kill the *Mtb* in broth culture (Supplementary Fig. 1). It may be that *Mtb* rapidly acquires resistance to D29, leading to the replacement of the susceptible *Mtb* in liquid cultures. Therefore, we conclude that the *Mtb* killing ability of D29 in agar plates does not necessarily correlate with the *Mtb* killing ability in liquid culture.

We further confirmed this outcome using a GFP-expressing H37Rv strain, H37Rv-GFP, in a liquid culture condition as described above. Due to the highly efficient expression of GFP gene, the live bacteria are visible under fluorescent microscopy. The microscopy images of the day 9 cultures clearly showed that there was no GFP expression in the wells with H37Rv-GFP co-cultured with DS6A, whereas H37Rv-GFP in the wells co-cultured with other phages remain fluorescent, further demonstrating the highly effective *Mtb*-killing ability of DS6A (Fig. 1F).

### Phage DS6A can efficiently eliminate *Mtb* H37Rv in primary human macrophages
To test whether the phages can kill *Mtb* in infected primary human macrophages, which is critical for therapeutic effectiveness in clinical settings, we isolated peripheral blood mononuclear cells (PBMCs) from the blood of a healthy donor and isolated the hCD14+ monocytes using microbeads to above 95% purity as determined by flow cytometry analysis (Supplementary Fig. 2). The purified CD14+ cells were differentiated into macrophages by incubation of the cells with hM-CSF, hGM-CSF, and hIL-4 for 5 days. The macrophages were then infected with H37Rv at an MOI of 1, and treated with different phages at an MOI of 10. On days 5 and 10, we plated the macrophage lysates on 7H10 agar plates to determine the CFUs (Fig. 2A). The results showed no significant reduction in *Mtb* CFUs for all phage-treated groups on day 5 when compared to the control *Mtb*-infected macrophages without phage treatment. However, on day 10, phage DS6A completely eradicated the bacilli from the infected macrophages, while other phages still failed to show any reduction in *Mtb* growth in macrophages (Fig. 2B, C).

We further tested whether the phages can also kill *Mtb* in infected macrophages from different donors. For this purpose, we isolated monocytes from four healthy donors, and repeated the above experiment in these four samples using phage DS6A, and phage D29 was used as a negative control. We also wanted to see whether *Mtb* could be killed in a shorter time rather than ten days, so we set the sampling time points to days 4 and 7. The results showed that all four donors displayed the same pattern in which *Mtb* in the infected macrophages were killed within seven days by phage DS6A, but not in the control phage treatment (Fig. 2D, E). Interestingly, some of the phage-treated cultures had a significantly higher bacillary load than the control bacteria cultures (Fig. 2B, D), which we address further in the discussion section.

To ensure that the phage DS6A can kill the intracellular *Mtb*, we infected the macrophages with H37R-GFP at an MOI of 1, and then treated them with phage DS6A. Infected macrophages without phage served as a positive control, and H37Rv-GFP uninfected macrophages were used as a negative control. We confirmed that *Mtb* bacilli entered the macrophages at 4 h post-infection using fluorescence microscopy (Supplementary Fig. 3). We then observed the cells at 7 days post-infection (dpi) using confocal miacroscopy. We found that at 7 dpi, *Mtb*-infected macrophages without phage treatment showed 20-30% GFP-positive signals (Fig. 2F, middle row), while the *Mtb*-infected macrophages treated with phage were completely void of GFP-positivity (Fig. 2F, bottom row). This result is consistant with the CFU result shown in Fig. 2B–E, further lending support that phage

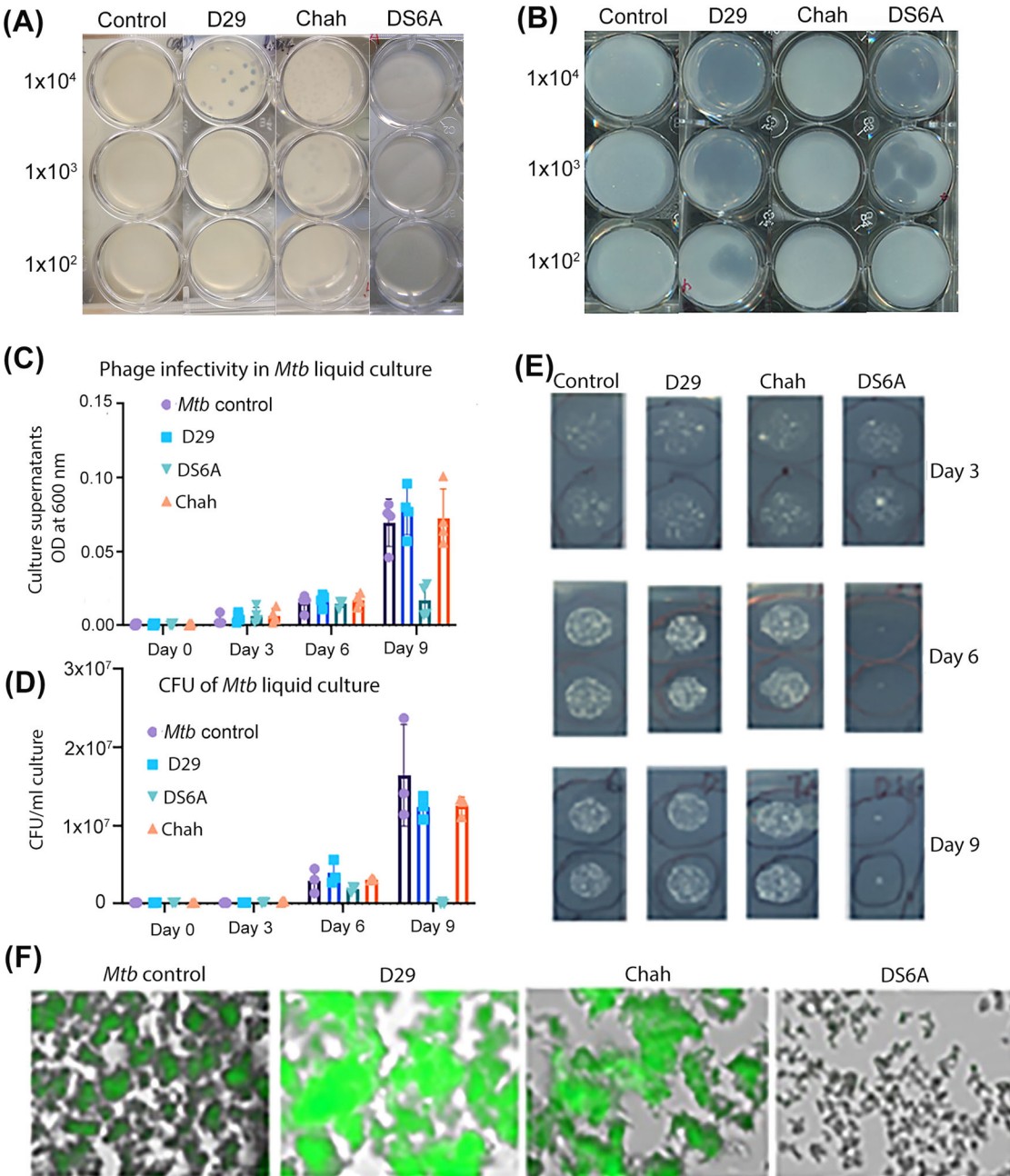

**Fig. 1 | Phage infectivity of *M. smegmatis* and *Mtb* H37Rv in solid agar plates, and in *Mtb* liquid culture.** Serial dilutions ($10^2$–$10^4$ pfu) of different strains of bacteriophages were mixed with $1 \times 10^5$ CFUs *M. smegmatis* (**A**) or H37Rv (**B**) in 7H9 medium and incubated at 37 °C with shaking for 1 h. The infection cultures were then mixed with 0.8% top agar and spread on 12-well 7H10 agar plates supplemented with 10% OADC enrichment. **C–F**: H37Rv ($1 \times 10^5$ CFUs) were infected with various bacteriophages at an MOI of 1 for 1 h, then inoculated into 20 mL 7H9 media supplemented with 10% ADC and incubated at 37 °C for 9 days. The cultures were sampled at days 3, 6, and 9 post infection by plating 10 μL of 10-fold serial dilutions of on the 7H10 agar plates and cultured at 37 °C for CFU determination. **C**: $OD_{600}$ of sampled liquid cultures was measured at different time points as indicated. **D**: Statistics of the CFUs of each sampled liquid culture sampled at various time points. **E**: Titering assay of phage-*Mtb* liquid culture sampled at different time points. The data are representative results of two independent experiments. **F**: Phage infection of H37Rv that expresses a GFP reporter gene. An MOI of 1 was used to infect H37Rv-GFP. GFP expression was pictured under fluorescence microscopy after nine days of phage infection.

DS6A can enter the macrophages and eliminate the intracellular growth of *Mtb*.

### *Mtb* infection of humanized NSG-SGM3 mice
Compared to the commonly used Hu-PBL and BLT mouse models, the humanized NSG-SGM3 mouse model has several advantages: (1) Humanized NSG-SGM3 mice exhibit improved reconstitution with an increased general population of human immune cells (hCD45 + ); (2) Cytokine expression supports human myeloid cell differentiation and maturation; thus the functional macrophages and dendritic cells are significantly elevated in the bone marrow compared with NSG recipients[28,31]; (3) There is a significant increase of regulatory T cells (Treg)[31], which are important in regulating *Mtb* pathogenesis[36,37]; (4) B cells can be developed to mature phenotypes with the ability of class-switching[27], which is important to evaluate the IgM and IgG responses against phages; (5) They are relatively easy to generate by intravenous injection of human HSC into irradiated adult mice for humanization.

**Fig. 2 | Phage DS6A eliminates *Mtb* in infected primary human macrophages. A** Experimental procedures of the assay. **B, C:** Half million macrophages were infected with H37Rv at an MOI of 1. After 4 h of *Mtb* infection, three different phages ($5 \times 10^6$, MOI of 10) were applied to the *Mtb*-infected primary human macrophages, and cultured at 37 °C. The cells were sampled at days 0, 5 and 10 to determine the bacillary load for the evaluation the *Mtb*-killing ability of the phages by plating the ultrasound-broken cells on 7H10 plates (supplemented with 10% OADC) and the CFUs were counted on Day 8. **B** shows the statistics of the CFUs of all phages, and (**C**) shows the representative pictures (duplicated) of the 7H10 plates taken at day 5 and day 10. **D, E:** Testing the *Mtb*-killing ability of phage DS6A in *Mtb*-infected macrophages derived from four different healthy donors, and phage D29 was used as a negative control. The infected macrophages were sampled at different time points, sonicated, and plated on 7H10 agar plates using different dilutions to titer the bacillary load. **D** shows the statistics of five donors, and (**E**) shows the representative pictures taken from 10x diluted plates for days 4 and 7 cultures. **F:** Confocal macroscopy images show the *Mtb*-killing efficacy of phage DS6A in H37Rv-GFP infected macrophages. Cellmask plasma membrane stain (red) was used to localize the cell membrane, GFP (green) was used for tracking the *Mtb* bacilli, and DAPI (blue) was used to stain the nuclei of the macrophages. The red arrows show the *Mtb* in the macrophages. Unpaired student T-tests were used to analyze the differences between groups in Figure **D**. *n* = 5 independent donors. All statistical data are represented as mean ± SEM. Statistical significance was defined as $*P \le 0.05$, $**P \le 0.01$, and $***P \le 0.001$.

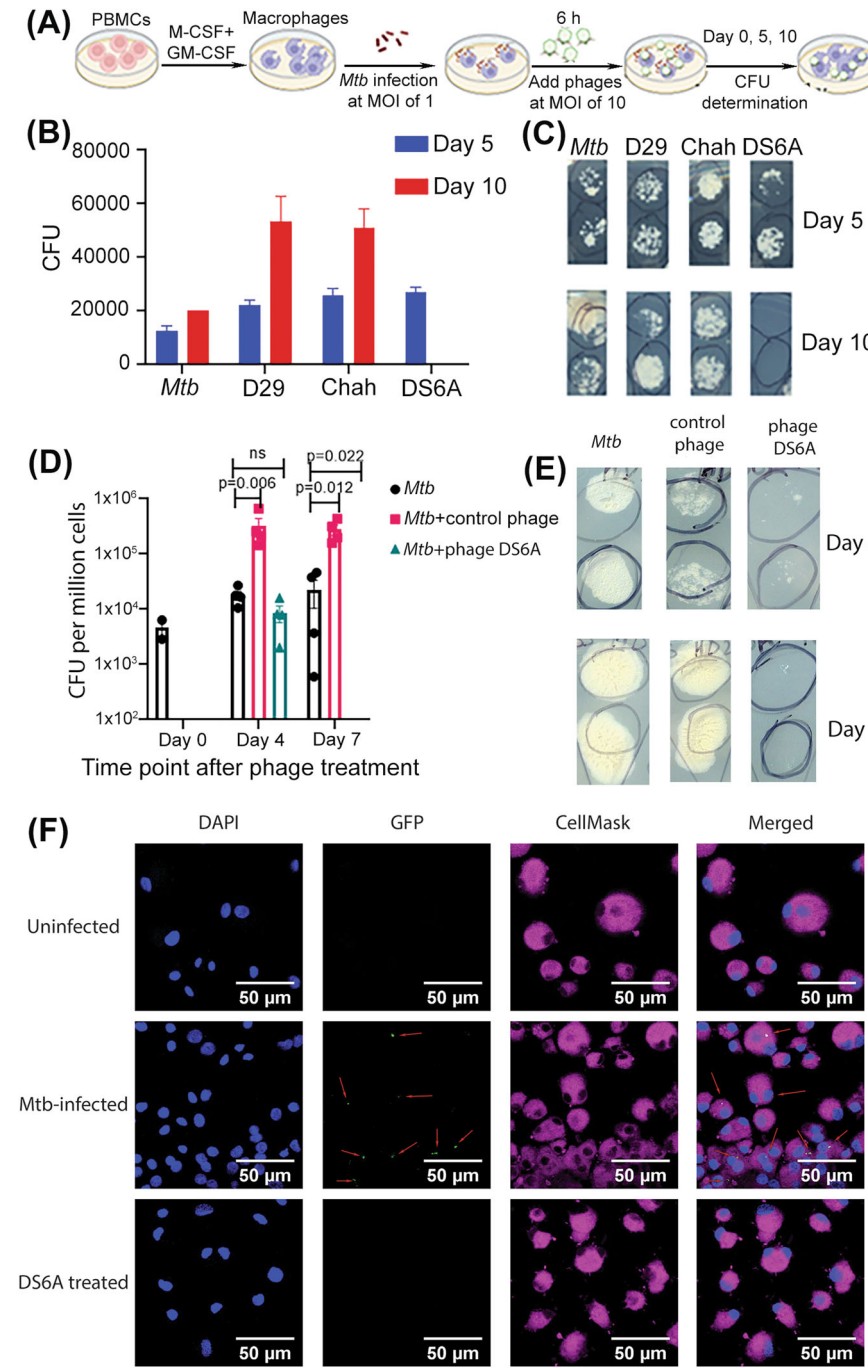

We obtained NSG-SGM3 mice from the Jackson Laboratory and have bred them in-house. For humanization, 4–6-week-old NSG-SGM3 mice were irradiated at 150 cGy and then infused with human CD34 + HSC. Ten to fifteen weeks after HSC injection, the animals developed a full complement of human immune cell types, including CD4+ and CD8 + T cells, B cells, myeloid cells, and natural killer (NK) cells (Fig. 3A, Gating strategy is shown in Supplementary Fig. 4).

To establish humanized NSG-SGM3 mouse model of *Mtb* infection, we infected the humanized NSG-SGM3 mice with H37Rv via a Madison chamber to deposit about 50–100 CFUs per mouse lung. At 25 days post-infection, we sacrificed the mice and analyzed the immune cell profile of lung and spleen homogenates. Additionally, we determined *Mtb* growth in lungs and spleens. The results showed that, when compared with the PBMCs analysis before infection, the spleen immune cells underwent substantial changes after infection, and the CD8 + T cells and myeloid cells

were significantly expanded (Fig. 3B). Meanwhile, in both the lung and spleen homogenates, we detected a high number of bacilli (Fig. 3C), indicating that the mice established *Mtb* reservoirs not only in the lungs but also systemically.

### Phage DS6A improved pulmonary function and eradicated splenic infection in *Mtb*-infected humanized mice

To investigate if phage DS6A can eradicate *Mtb* in vivo, we performed an animal experiment in the humanized NSG-SGM3 mouse model of tuberculosis (Fig. 4A). We infected the humanized NSG-SGM3 mice with low dose aerosolized H37Rv, as confirmed by an average ~100 bacilli per lung deposited at day one post-infection after sacrificing three animals to evaluate the CFU in the lung homogenates (Fig. 4C). From day 3 of *Mtb* infection, we treated the mice every other day with DS6A ($10^{11}$ phage particles/dose) for a total of 10 doses via intravenous administration.

**Article**

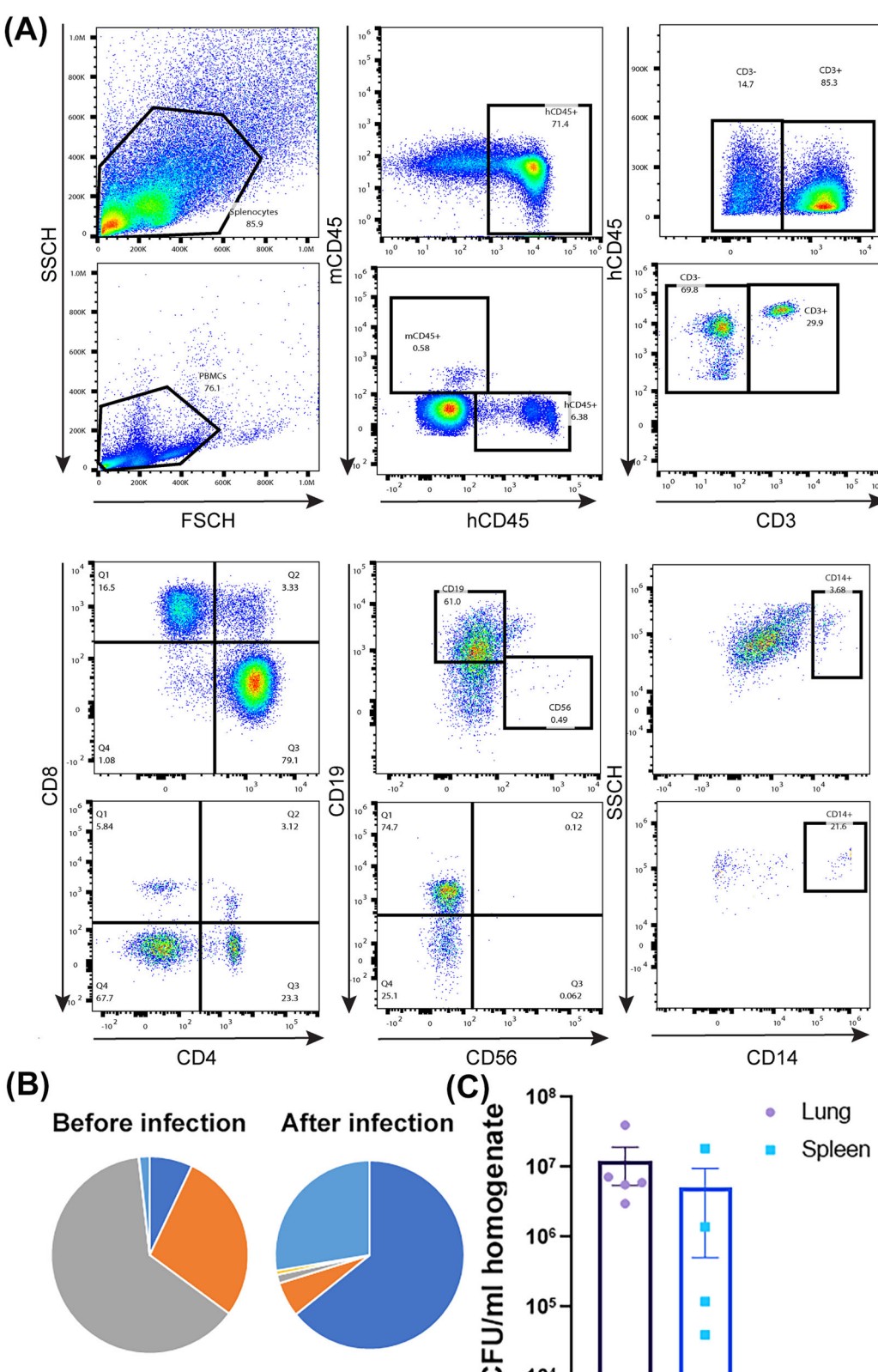

**Fig. 3 | *Mtb*-infection of humanized NSG-SGM3 mice. A** Reconstitution of human immune cells in splenocytes (upper panels) and PBMC (lower panels) from humanized NSG-SGM3 mice. NSG-SGM3 mice were irradiated and intravenously injected with $2 \times 10^5$ hCD34+ HSC. After 10–12 weeks, reconstitution of human T cells, B cells, myeloid cells, and natural killer (NK) cells within human CD45-gated population was analyzed by flow cytometry. **B** Statistics of different immune cells before and after infection. **C** Five mice were sacrificed to check *Mtb* infection in lungs and spleens by determining the organ CFUs by plating serially diluted organ homogenates on 7H10 agar plates as described above.

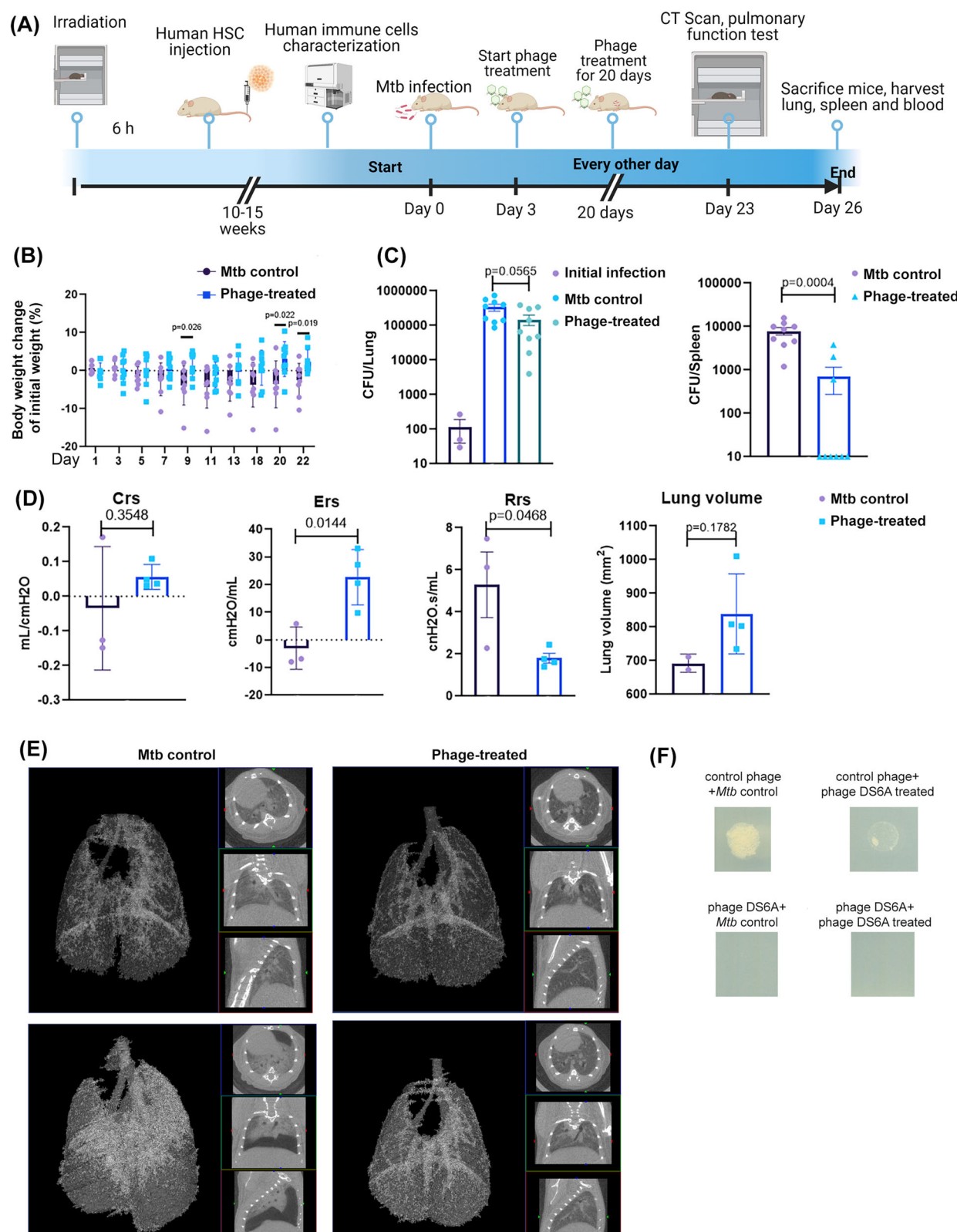

After the treatments, we determined the growth of *Mtb* in mouse lungs and spleens. We also performed pulmonary function tests (PFTs) and evaluated overall mouse health by monitoring mouse body weight changes in association with changes in the bacillary burden of the mice. The results showed that DS6A-treated *Mtb*-infected mice gained significantly more weight than control *Mtb*-infected mice (Fig. 4B). The PFTs also showed that two of the four parameters (Ers and Rrs) tested in the phage-treated group had significantly improved, with less respiratory resistance and better elasticity, demonstrating improved pulmonary function after treatments (Fig. 4D). The CT scan results showed that there were fewer high-density areas with much cleaner lungs in the phage-treated group of mice, which indicates less inflammation and other pathological changes (Fig. 4E).

**Fig. 4 | Phage DS6A is effective at eradicating *Mtb* in humanized NSG-SGM3 mice. A** Schematic presentation of the animal experiment procedures. H37Rv aerosol infected humanized mice were treated three days after infection with $1 \times 10^{11}$ pfu of phage DS6A per mouse or equal volume of sterile PBS every other day for eight times. **B** The mouse body weights were monitored over time, and the body weight change over the initial body weight of the mice are shown as percentages ($n = 9$ for each group). **C** The growth of H37Rv in the mice lung and spleen homogenates together with initial infection dose (left panel in purple) are shown ($n = 3$ for initial infection dose determination; $n = 9$ for *Mtb* control and phage-treated group). **D** Before termination of the experiment, the mice were subjected to pulmonary function testing. The measurements of elastance, compliance, total lung resistance, and total lung volume were collected, and the statistics are shown ($n = 3$ for *Mtb* control group, $n = 4$ for phage-treated group). **E** CT scans were performed for each mouse, and two representative figures for each each group show the *Mtb*-infected control mouse and phage-treated mouse lungs, respectively. The left panel

of each mouse CT scan figure shows the 3D image, the white areas represent the high-density scan (e.g., tissues), while the black areas represent low density scan (e.g., air). The three small figures in each mouse scan show different angles of scan results. **F** The colonies from the plates that spread with lung or spleen homogenates of the H37Rv infected control or phage-treated mice were picked and mixed with $1 \times 10^6$ pfu of phage DS6A or a negative control phage (phage Chah) in 10 μl 7H9 media and incubated at 37 °C for 1 h. Then the infection cultures were pipetted onto the 7H10 agar plates OADC and incubated at 37 °C for determination of CFUs. The animal experiments were performed four times, and the 3rd and 4th experiments were shown. Except the pulmonary functions were tested once (3rd animal experiment, $n = 3$ for Mtb control group, and $n = 4$ for phage DS6A-treated group), the other data are shown as the combined results from the 3rd and 4th experiments ($n = 9$ for each group). All the statistics are shown as mean ± SEM. Unpaired Student T-test was used to analyze the differences between groups. Statistical significance was defined as *$P \leq 0.05$, **$P \leq 0.01$, and ***$P \leq 0.001$.

Furthermore, the lung volume of the phage-treated mice was larger than untreated *Mtb* control mice, even though the difference didn't reach significance ($p = 0.1782$) (Fig. 4D). The CFU counts showed that the bacilli were completely eradicated in the spleens in six of nine mice of the phage-treated group, and the average bacillary load in the spleen was significantly lower than in the *Mtb*-infected control group (Fig. 4C). Moreover, the lung bacillary load showed three-times higher in *Mtb*-infected control mice than in phage-treated mice, while the difference didn't reach significance (Fig. 4C). These results suggest that, after intravenous administration, phage DS6A has the capacity to kill the replicating *Mtb* within/or outside the lung that are disseminated to the blood circulation.

To explain why the phage DS6A performed better in killing *Mtb* bacilli in spleens than in lungs, we determined the phage genome copies in the lung and spleen homogenates. The quantitative PCR result showed that the spleen had three times more phage genome copies than the lungs (Supplementary Fig. 5), suggesting that the effectiveness in tissues may be positively correlated to the phage copies, and that phages intravenously administered was preferentially distributed to the filter organs such as spleen and liver more than to the lungs[38], which may explain the organ-dependent differences in *Mtb* eradication in humanized mice.

Given that phage DS6A could not wholly eradicate the spleen bacteria in several mice, we wondered whether phage resistance developed in the bacteria within these mice. For this purpose, we picked colonies from the plates that spread the lung and spleen homogenates of phage-treated mice, and then mixed them with phage DS6A to see if the phage could kill the bacteria in vitro. The phage Chah was used as a negative control. The results showed that the negative control phage could not kill bacteria colonies picked from DS6A-treated or untreated mice, while the DS6A could kill both (Fig. 4F), indicating that no phage resistance developed during the 20-day phage treatment period.

**Phage therapy elicits antibody responses in humanized NSG-SGM3 mice without affecting the function of immune cells**

There is always a concern that the immune responses, especially antibody responses, will stymie the effect of phage therapy. Therefore, we determined the phage DS6A-specific antibody responses in experimental mouse sera by ELISA. We detected a significantly higher level of human IgM in phage DS6A-treated mouse sera (Fig. 5B). However, we could only detect weak IgA and IgG responses in these mice (Fig. 5A, C). We further tested if there are any neutralizing antibody responses that may mitigate the phage treatment efficacy, but we could not detect any significant neutralizing antibody responses in any serum dilutions of all phage-treated humanized mice (Supplementary Fig. 6).

We further tested whether phage therapy could affect the cytokine profiles associated with reduced *Mtb* growth and improved lung function due to the phage treatment. We determined the levels of 27 cytokines including cytokines of human lymphocytes and myeloid cells in the lung and spleen homogenates of the mice by a multiplex cytokine assay (Supplementary Fig. 7). Overall, the cytokine results showed that

the cytokine levels in the spleen were higher than that of the lungs, and that the phage treatment induced significantly fewer myeloid cell-derived cytokines than observed in the control *Mtb*-infected mice. This observation is consistent with reduced bacilli burden in the mice treated with phage. We did not see significantly elevated levels of T cell cytokines in the mice after four weeks of *Mtb* infection, consistent with significantly increased CD14+ monocytes, but not lymphocytes, in the humanized mice after *Mtb* infection.

## Discussion

The development of alternative treatments to antibiotics for *Mtb* infection is an urgent task in the era of antibiotic resistance. In this study, we have shown that bacteriophages D29 and DS6A can efficiently kill *Mtb* bacilli on 7H10 agar plates, while only phage DS6A significantly killed *Mtb* in liquid cultures. However, only phage DS6A was able to kill *Mtb* bacilli in primary human macrophages. In addition, we showed that phage DS6A could effectively eradicate *Mtb* bacilli in spleens and reduced the bacilli load in the lungs in a humanized mouse model of *Mtb* infection. Thus, phage DS6A appears to have the potential to be developed as an effective therapeutic for *Mtb* treatment.

Phage therapy has shown great potential in controlling some intransigent drug-resistant bacterial infections. Compared to typical antibiotic treatment, phage therapy has advantages such as: host specificity, low toxicity, the ability to lyse the bacteria, sustained therapeutic levels in situ, due to phage replication, and a narrower potential for inducing resistance, all of which render it an attractive treatment strategy in the era of antibiotic resistance. Several phage strains have shown potential in eliminating *Mtb* infection[20]. Unfortunately, there has been little progress in *Mtb* phage therapy, with only two studies published to date that reported on the use of phage therapy for the treatment of *Mtb* in guinea pigs, with limited success[39,40]. Therefore, our study showing the effectiveness of phage therapy in treating *Mtb* infections of clinically relevant humanized mice represents a significant advancement in this field.

Humanized mouse models provide versatile tools to study various infectious diseases. For *Mtb* infection, the myeloid cell lineage is especially important because the macrophages are the main targets of the bacilli. In most NSG-based humanized mice, myeloid cell differentiation is not efficient, and the macrophages are not mature, thus the physiological processes such as phagocytosis and endocytosis, which may be involved in *Mtb* infection and bacteriophage uptake, may not be completed as properly as they are in humans. However, due to the transgenic expression of the three human cytokine genes, the myeloid cells were well-differentiated in NSG-SGM3 humanized mice that we have used in this study. After *Mtb* infection, the myeloid cells were successfully propagated (Fig. 3B). Moreover, after phage treatments, the phage was shown to be capable of killing *Mtb* in human macrophages engrafted in the humanized mice, suggesting that NSG-SGM3 mice can recapitulate human *Mtb* infections, and thus NSG-SGM3 mice are an appropriate and clinically relevant animal model for *Mtb* phage therapy.

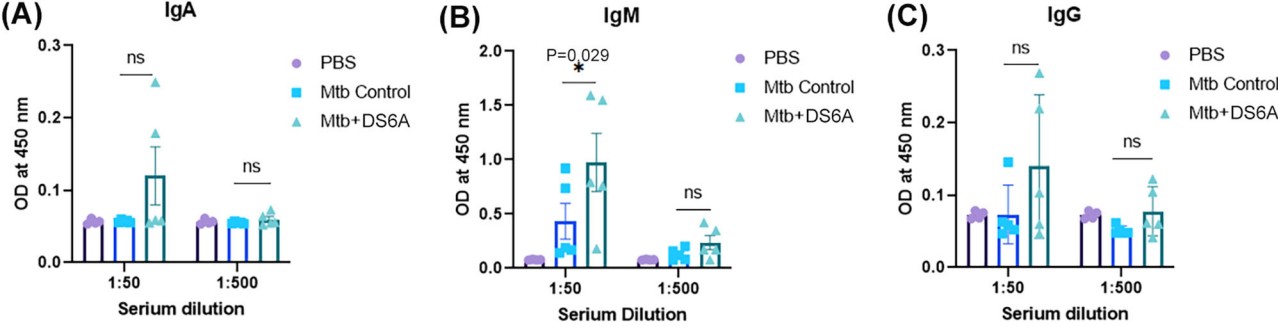

**Fig. 5 | Humanized NSG-SGM3 mice developed antibody responses against phage DS6A.** ELISA was performed to determine the phage DS6A-specific human IgA (**A**), IgM (**B**), and IgG (**C**) titers in sera of the humanized mice infected with H37Rv and treated with phage DS6A or PBS buffer control. The experiment was repeated once, and the results from one experiment with five mice each group are shown. All the statistics are shown as mean ± SEM. Unpaired Student's T-test was used to analyze the differences between groups. Statistical significance was defined as *$P ≤ 0.05$, **$P ≤ 0.01$, and ***$P ≤ 0.001$.

Although D29 and DS6A showed the ability to kill *Mtb* on bacterial lawns, only DS6A efficiently killed *Mtb* in liquid culture and in human primary macrophages. Additionally, the proven capacity of DS6A to reduce bacterial load in humanized mice suggests that DS6A is the most appropriate phage therapeutic for *Mtb* infections. The reasons why D29 can kill H37Rv efficiently on bacterial lawns, but not in liquid culture, are not fully clear (Fig. 1). The most likely explanation is that *Mtb* readily acquires resistance to D29, but not to DS6A. When inoculated in liquid culture at an MOI = 1, it may be that at least some *Mtb* cells are genetically incapable of being infected by D29, thus these resistant *Mtb* cells eventually dominate the culture. Additionally, it appears that D29 cannot easily break this resistance. Alternatively, it is possible that D29 is capable of lysogenizing *Mtb*, but the available evidence suggests that this is not the case. First, D29 does not form turbid plaques on host lawns suggesting that, if it is capable of lysogeny, it does so at a very low rate[41] (Fig. 1). Moreover, although D29 does have an intact attP-integration system, no repressor gene required for the maintenance of lysogeny and superinfection has been identified[41]. While it is possible that D29 possesses a cryptic or unrecognizable repressor, a genome analysis suggests that it does not. D29 is highly homologous to the mycobacteriophage L5, but has a 3.6 kb deletion relative to L5 that removes a part of its genome that corresponds to L5's repressor gene[41]. Nevertheless, these results indicate that liquid cultures, rather than bacteria lawns, appears to be a better prediction model for the in vivo efficacy of *Mtb* phage therapy.

Of note, we also observed that once the phage cannot kill *Mtb* in liquid culturing condition, it sometimes will stimulate *Mtb* growth, as shown in Fig. 2. The observation that, in some cultures, bacterial load is higher in the presence of phages D29 and Chah but not DS6A, than in phage-free controls (Fig. 2B, D) is puzzling. We postulate that the reaction between the phages (D29 and Chah) and macrophages may interfere with bacteria growth, or this may be due to the selective pressure caused by phages that results in the fast propagation of phage-resistant cells[42]. However, the exact mechanism is worth investigating in the future study.

Nevertheless, our experiments provide clear evidence that phage DS6A was effective in killing *Mtb* in the spleen via the intravenous route. However, the bacterial clearance in the lungs was not as effective as in spleens. This discrepancy is possibly due to the administration route, as this is one of the most important factors that affects in vivo efficacy of phage therapy[43]. Through the intravenous route, phages can easily reach the spleen and liver through circulation as blood filtering organs, but the lungs are much more difficult to reach. A previous study showed that the intravenous delivery of a pneumonia bacteria-specific phage led to a 2-log lower prevalence (<100 times) in the lungs when compared to an intratracheal delivery route[43]. Additionally, phages need to penetrate the lung epithelial barrier[44] in order to access the alveolar macrophages that are parasitized by *Mtb*, thus further increasing the difficulty of efficient intravenous phage administration. We note that although the lung bacilli burden between the *Mtb* control mice and the phage-treated mice was not significant, the *P* value (0.0565) is very close

to significance. We reason that the less bacilli burden in phage-treated group (~8-fold less than in *Mtb* control group) contributed to the improved lung function. However, we cannot exclude any other reason for the improved lung function.

To test the hypothesis that phage delivery to the lungs was restricted, we compared phage copy number in spleens and in lungs, and the results indicated there are more phage copies in spleen than that in lungs. It is worth noting that our original dose contained $10^{11}$ phages per dose, while we only detected thousands of copies at the termination time point. This discrepancy may be due to the pharmacodynamical changes of the phages, as the phage's half-life might be as low as 2.2 h and some studies have reported that phages are undetectable after 36 h of injection[45,46]. Even though we need to further investigate the pharmacokinetics of the phage DS6A, our qPCR result offered a reasonable explanation of why the intravenous administration of phage is more efficient in spleens than in lungs. Nevertheless, our results suggest that optimization of administration will be helpful to improve lung bacterial clearance (e.g., via intratracheal route or nebulizer delivery, etc.) as tuberculosis, in majority of cases, are infections of the lungs.

Phage resistance has always been a concern as it may dampen the therapeutic efficacy of phage therapy. In our study, it seems not to be a case for phage DS6A within the treatment period (Fig. 4f). Similarly, in a recent compassionate clinical use of phage therapy for the treatment of 20 patients with Nontuberculous Mycobacterium infection, no phage resistance was found in any of the patients[47]. However, in the clinical settings of *Mtb* phage therapy, the treatment time may be as long as several months, and *Mtb*-resistant strains may still appear. Therefore, screening more effective bacteriophage strains to form a phage cocktail would be essential to overcome phage resistance by *Mtb*.

Moreover, our study showed that significant IgM but weaker IgA and IgG responses were induced after 20 days of phage treatment (Fig. 5). This may be because the short-term administration of phage may lead to IgM as dominant antibodies, and with the prolonged administration, the IgG response will become dominant. Given that phage treatment still improved disease progression in a human clinical case of *M. chelonae* despite a significant anti-phage antibody response[18], it may be that are not a great concern. Indeed, in our study, the antibody responses did not adversely affect the efficacy of phage therapy to eradicate *Mtb* in the spleen. This may be because our treatment regimen was not deployed long enough to enable to the development of sufficient quantity and quality of antibody responses to counteract the high dose of phages we administered. However, in clinical settings where a long treatment regimen might be needed, a high dose might be warranted to reduce the effects of antibody responses to the phages.

Taken together, our study showed that bacteriophage DS6A could effectively eliminate *Mtb* in agar plates, in liquid culture and in infected macrophages. In addition, using a humanized mouse model, we have shown the efficacy of DS6A to eradicate bacteria from infected organs in live animals, even though the biodistribution of the administered phages

remains a challenge to achieve complete elimination. Therefore, our study demonstrated the feasibility of developing phage therapy as an effective therapeutic against *Mtb* infection. Our current study focused on the efficacy in human macrophages and in vivo in humanized mice, while there are a lot of questions that remain uninvestigated, such as how the phages penetrate into the macrophages, whether and how the phages penetrate the granulomatous environment to kill the *Mtb* bacteria, and how the phages are distributed during the in vivo treatments, etc. The investigations into these detailed mechanisms are our ongoing and planned future studies.

## Materials and methods
### Regulatory and ethical statement
Healthy donors were recruited for the collection of blood samples from the employees and students at the University of Texas Health Science Center at Tyler (UTHSCT) following the protocol approved by the Institutional Review Board (IRB) ethics committee of the UTHSCT (Protocol #2022-003). All animal procedures were approved by the UTHSCT Institutional Animal Care and Use Committee (IACUC) (Protocol #691).

### Bacterial strains, phage strains, media, and culture conditions
Bacteriophage DS6A was purchased from the ATCC (Cat# 25618-B2), and phages D29 and Chah were prepared at Queens College, The City University of New York (and originally obtained from Dr. Graham Hatfull's lab at the University of Pittsburgh). *M. smegmatis* mc2 155 was purchased from the ATCC (Cat# 700084). *Mtb* H37Rv and *Mtb* H37Rv-GFP were prepared in our laboratories at UTHSCT. Except for DS6A, all mycobacteriophages were amplified in *M. smegmatis*; DS6A was amplified in *Mtb* H37Rv in the BSL-3 Laboratory at UTHSCT.

### Amplification and quantification of bacteriophages
Bacteriophages, except DS6A, were amplified in *M. smegmatis*. *M. smegmatis* was streaked on 7H10 agar plate supplemented with 10% Middlebrook Oleic Albumin Dextrose Catalase (OADC), 1 mM $CaCl_2$, 10 μg/mL of carbenicillin, and 100 μg/mL of cycloheximide and cultured at 37 °C for about 3–4 days for single colony growth. A single colony of *M. smegmatis* was cultured in 7H9 media supplemented with 10% Middlebrook Albumin Dextrose Catalase enrichment supplement (ADC), 1 mM $CaCl_2$, 10 μg/mL of carbenicillin, 100 μg/mL of cycloheximide, and 0.25% Tween-80 at 37 °C for 3–4 days until the OD at 600 nm was over 2.0. A hundred μLs of *M. smegmatis* from the above culture was inoculated into 5 mL of 7H9 media supplemented with 10% ADC, 1 mM $CaCl_2$, 10 μg/mL of carbenicillin, 100 μg/mL of cycloheximide without Tween-80, and incubated overnight with shaking at 200 rpm at 37 °C. Next day, $10^8$ pfu of bacteriophages with 100 μL of *M. smegmatis* were mixed and incubated at 37 °C for 10 min, then the phage-bacilli mixture was added to 7 mL of top agar at 60 °C, and immediately applied onto 150-mm 7H10 agar plates. The phage-bacilli agar plates were incubated at 37 °C overnight, then the bacilli were collected in 10 mL of phage buffer (10 mM Tris, pH 7.5, 10 mM MgSO4, 0.4% NaCl) and stored at 4 °C overnight. After centrifugation, the supernatants were collected and filtered through a 0.22 μM PVDF filter, and the phages were stored at −80 °C until use.

*Mtb* was cultured in 7H9 with 10% ADC following the standard *Mtb* culture procedures. *Mtb* single cell suspensions at $10^8$ cells per ml were infected with phage DS6A using an MOI of 1 for 10 min at 37 °C. The infection mixture was mixed with 7 mL of top agar (0.8% agarose in 7H9 with 10% ADC) and applied onto 150 mm 7H10 agar plate (supplemented with 10% OADC, 1 mL of $CaCl_2$). The plates were incubated at 37 °C for ~8 days. After plaques formed, the phage DS6A was collected in phage buffer and filtered through a 0.22 μM PVDF filter.

Bacteriophages' titers were determined by plaque assays by infection of H37Rv with phage DS6A or *M. smegmatis* with other phages, respectively, after serially diluted the phages. Alternatively, quantitative PCR was also used to quantify the viral genomes.

### Testing *Mtb*-killing ability on 7H10 agar plates
To test whether the bacteriophages can kill *Mtb* on the solid agar plate culture, we performed the plaque assays. Briefly, $10^4$, $10^3$, $10^2$ pfu of D29, Chah (for both the titers based on *M. smegmatis*), and DS6A (the titer based on *Mtb*) were mixed with $1 \times 10^5$ *Mtb* bacilli and incubated at 37 °C for 1 h with shaking. The *Mtb*-phage mixture was then mixed with 500 μL of 0.8% top agar in 7H9 with 10% ADC and applied to 12-well 7H10 agar plates. The plates were incubated at 37 °C for ~7–8 days until the plaques developed.

### Testing *Mtb*-killing ability in 7H9 liquid culture
The *Mtb* killing activities of the bacteriophages were determined by infecting $1 \times 10^5$ H37Rv in 1 mL 7H9 media with various bacteriophages at an MOI of 1 for 10 min, then inoculated into 20 mL 7H9 media supplemented with 10% ADC and incubated at 37 °C for 9 days. At days 3, 6, and 9, 10 μL of 10-fold serial dilutions of the culture were plated on 7H10 agar plates, and CFU were counted after cultured at 37 °C for 2–3 weeks.

### To test the phage D29 *Mtb*-killing capacity in 7H9 liquid culture using different MOIs
H37Rv ($1 \times 10^5$ CFU in 1 mL 7H9 media) were inoculated with D29 phage at an MOI of 1, 10 and 100, respectively. Afterwards, the mixture was inoculated in 20 mL of *Mtb* growth media (7H9 supplemented with 10% ADC), followed by incubation at 37 °C. One mililiter of bacterial culture was recovered every 3 days and 10-fold serial dilutions were plated in 7H10 agar plates supplemented with 10% OADC. Following incubation at 37 °C for 2 weeks, CFU were counted.

### Flow cytometry
Cells from one well were collected and stained for macrophage markers. The antibodies used were APC-conjugated CD11b (cat# 301350), FITC-conjugated CD14 (cat# 325604), BV711-conjugated CD16 (cat# 302044). The isotype controls included APC-mouse IgG1 (cat# 400142), FITC-mouse IgG1 (cat# 400108), BV711-mouse IgG1 (cat# 400168). All antibodies were purchased from BioLegend. Briefly, the cells were washed with PBS and incubated with the antibodies at room temperature for 15 min, avoiding light. The cells were then washed and resuspended with a buffer of 0.5% BSA/PBS + 3 mM EDTA prior to analysis with Attune NXT (ThermoFisher). Compensation was performed using UltraComp eBeads Plus (01-3333-42, Invitrogen). The data were then analyzed with FlowJo version 10.6.1, and the graphs were created with GraphPad Prism 8.

### Testing the *Mtb*-killing ability of the bacteriophages in macrophages
Blood was drawn from five healthy donors and PBMCs were isolated using Ficoll-Paque Plus (17-1440-02, GE Healthcare, Danderyd, Sweden) gradient centrifugation. Then, CD14+ cells were isolated from the PBMCs using human CD14+ micro beads conjugated with anti-human CD14 mAbs (130-118-906, Miltenyi Biotec, Bergisch Gladbach, Germany) LC columns (130-042-401, Miltenyi Biotec, Bergisch Gladbach, Germany) following the manufacturing instructions. The purity of the cells was checked with flow cytometry.

Half a million cells were seeded in each well of 24-well plates containing 1.5 mL of RPMI-1640 supplemented with 10% FBS and 1% penicillin + streptomycin. The cells were then incubated in the presence of 50 ng/mL of GM-CSF (02532, Stem Cells, Vancouver, Canada) and 50 ng/mL of M-CSF (216-MC, R&D Systems, Minneapolis, USA) at 37 °C and 5% $CO_2$ for differentiation. Fresh cytokines were replenished on day 3 once. On day 6, some cells were collected and stained for macrophage markers followed by flow cytometry analysis. The rest of the cells were washed with PBS and rested overnight in fresh medium without serum and antibiotics. The cells were then infected with H37Rv at an MOI of 1. Four hours post infection, the cells were washed with warm PBS thrice to remove extracellular bacilli. The cells were then incubated in fresh RPMI-1640 with 10% FBS with no antibiotics as controls and with either DS6A or D29 control phage at 37 °C.

The growth of H37Rv in macrophages were followed for different time points after infection by plating serially diluted cellular lysates on 7H10 agar plates as follows: After removing the culture medium, the cells were treated with ddH$_2$O for 10 minutes followed by 0.25% SDS in HBSS for 10 min. At the end, 20% BSA in PBS was added to stop the lysis. The content of each well was collected into a tube and sonicated three times with 33% amplitude for 10 s. After making dilutions, 10 μL of cellular lysate was plated on 7H10 agar plates in duplicates. The plates were incubated at 37 °C for 2–3 weeks and the bacterial colonies were counted for CFU determination.

**Fluorescence confocal microscopy**

PBMCs from a healthy donor (Stemcell Technologies, Vancouver, Canada) were thawed and monocytes were isolated using magnetic beads, as described above. Following isolation, $1 \times 10^6$ monocytes were seeded on each chamber of a 4-chamber slide (Celltreat scientific products, Pepperell, MA) and macrophage differentiation was carried out as previously described in the manuscript. After 6 days of differentiation, the media was replaced with fresh RPMI 1640 + 10% FBS without antibiotics and, the next day, macrophages were infected with GFP-expressing H37Rv at an MOI of 1. Following incubation at 37 °C for 4 h, the media was removed, and the cells were washed 3 times with PBS to remove extracellular bacteria. The infected macrophages were incubated with phage DS6A (at an MOI of 100) or PBS buffer for 4 h, then the cells were washed with PBS for three times to remove extracellular phages, and the infected macrophages were cultured in RPMI 1640 + 10% FBS in the absence of antibiotic and replenished with fresh media every 2 days. After 7 days of incubation, the media was removed and the cells were washed 3 times with PBS, followed by the addition of 1 ml/chamber of CellMask™ Deep red Plasma Membrane Stain (Invitrogen, Waltham, MA) diluted 1:1000 in PBS. The cells were incubated at 37 °C for 10 min and the membrane stain was removed. The cells were then fixed with 4% paraformaldehyde for 30 min at room temperature and permeabilized for 10 min using 0.1% (vol/vol) Triton X-100 (Fisher Scientific, Hampton, NH) diluted in PBS. The macrophages were blocked by incubation with 2% BSA in PBS at room temperature for 1 h. The chambers were then removed, and the slides were mounted using mounting medium with DAPI (Abcam, Cambridge, UK). Images were captured with a Zeiss LSM 510 confocal microscope (Zeiss, Hebron, KY), using the 63X oil objective lens. Fluorescent images in the blue (DAPI), green (GFP) and far red (Cellmask™) channels were acquired with the LSM ZEN 2009 software (Zeiss) and processed using the ImageJ software (NIH).

**Generation of humanized mice**

Humanized NSG-SGM3 mice (NSG mice transgenically expressing three human cytokine/chemokine genes IL-3, GM-CSF, and KITLG, RRID: IMSR_JAX:013062) were obtained from Jackson Laboratories and bred in UTHSCT vivarium. To increase scientific rigor, the experiments have been done in a blinded manner. To exclude sex bias, each experiment was considered to contain an equal number of male and female mice. For humanization of NSG-SGM3 mice, the mice were irradiated with a dose of 100 cGy per mouse and injected intravenously with 0.2 million of HSCs (purchased from Stemcell Technologies Inc.). After 8–10 weeks, the human cell reconstitution was confirmed by flow cytometry analysis of human immune cells in the mice by staining the PBMCs with fluorescent human antibodies specific for CD45, CD3, CD4, CD8, CD14, CD19, and CD56.

***Mtb*** **infection and phage treatment of humanized mice and other animal experiments**

Humanized mice were infected with a low dose of aerosolized H37Rv in a Madison chamber as previously described[48] by properly diluted H37Rv stocks in 10 mL normal saline to deposit about 100 CFUs per mouse lung. Bacterial inoculum dose was confirmed by the determination of Lung CFUs in three animals, euthanized 1 dpi. All animal experiments were done in a

blinded manner to increase scientific rigor. Wherever possible, each experiment contains equal numbers of male and female mice to exclude sex bias.

*Mtb*-infected mice were treated with $1 \times 10^{11}$ pfu phage DS6A per mouse or equal volumes of PBS as a control by intravenous administration every other day for a total of 10 treatments. The mice's body weight was measured every other day during the treatment. Three days after the last treatment, the mice were subjected to pulmonary function testing and CT scan as described previously[49]. Briefly, mice were anesthetized with a ketamine/xylazine mixture, and the anesthetized mice were cannulated with a sterile, 20-gauge intravenous cannula through the vocal cords into the trachea. The snapshot perturbation method was used to evaluate lung compliance by collecting the measurements of elastance, compliance, and total lung resistance for each mouse using the FlexiVent system (SCIREQ, Tempe, AZ). The flexiVent was set to a tidal volume of 30 mL/kg at a frequency of 150 breaths/min against 2–3 cm H$_2$O positive end-expiratory pressure, according to manufacturer's instructions.

After lung function testing, the mice were immediately subjected to CT scans for the measurements of lung volume. The Explore Locus Micro-CT Scanner (General Electric, GE Healthcare, Wauwatosa, WI) was used for CT imaging. CT scans were performed during full inspiration and at a resolution of 93 μm. Lung volumes were calculated from lung renditions collected at full inspiration. Microview software 2.2 (http://microview.sourceforge.net) was used to analyze lung volumes and render three-dimensional images. The mice were maintained under anesthesia using isofluorane throughout the pulmonary function testing.

At the end of testing, the mice were euthanized, and the lungs and spleens were collected and homogenized. The serially diluted lung and spleen homogenates were plated on 7H10 agar plates with OADC and incubated at 37 °C for 2–3 weeks for the determination of the organ CFUs.

**ELISA for detection of phage-specific antibodies**

Phage DS6A were coated in 96-well Nunc-Immuno plates (Thermofisher) with a concentration of $5 \times 10^9$ pfu/mL in coating buffer (Na2CO3, pH8.0) at 4 °C overnight. After washing with PBST (0.1% Tween-20 in PBS) four times, the plate was blocked with PBSM (5% skimmed milk in PBS) and followed by applying 10-fold serially diluted humanized mouse sera and incubated for 2 h at room temperature, then washed with PBST for 5 times. 100 mL of HRP-labeled secondary antibodies were applied to the plates and incubated at room temperature for 1 h. The secondary antibodies included: (1) goat anti-human IgG Fc (HRP) pre-adsorbed (catalog no. ab98624; Abcam); (2) goat anti-human IgA alpha chain (HRP) pre-adsorbed (catalog no. ab98558; Abcam); and (3) goat anti-human IgM mu chain (HRP) pre-adsorbed (catalog no. ab98549; Abcam). After washing away the unbound secondary antibodies, TMB substrate was added and the OD values at 450 nm were recorded using a BioTek Synergy 2 plate reader.

**Testing the phage DS6A-treated humanized mouse sera for the neutralizing ability against phage DS6A**

The neutralizing antibody response in serum samples from phage-treated humanized mice was evaluated following the protocol described previously[50,51] with modifications to account for the serum sample volume. Serum samples from phage treated mice were thawed and inactivated at 56 °C for 1 h in a water bath. Serial four-fold dilutions of serum were prepared and 225 μl of serum dilution were inoculated with 25 μl of DS6A lysate. The original titer of the phage used was $1 \times 10^6$ pfu/ml. Following incubation at 37 °C for 1 h, 200 μl of growing-phase *Mtb* (OD value of 0.5) were added to the mixture and incubated at 37 °C for an additional 10 min. Afterwards the mixture was added to 1.5 ml of melted top agar and poured in a six-well plate, previously coated with 7H10 agar, supplemented with 10% OADC. After 2 weeks of incubation at 37 °c, the plates were observed to assess plaque formation due to phage growth.

## Multiplex assay for cytokine profiling

Lung and spleen tissue samples from both experimental groups were homogenized in PBS to a final volume of 1.5 mL, using a 70 µm cell strainer (MTC Bio, Sayreville, NJ). The homogenate was passed through a 0.2 µm filter (Sigma-Aldrich, St. Louis, MO), and the flowthrough was used to evaluate the cytokine profile in these tissues. The samples were analyzed in duplicates using the Bio-Plex Pro™ Human Cytokine panel (Bio-Rad, Hercules, CA), according to the manufacturer's instructions. Briefly, 50 µL of filtered tissue homogenate were dispensed in a 96-well plate containing magnetic beads conjugated with antibodies for the detection of 27 different cytokines. After further incubation with detection antibodies and streptavidin-PE, the samples were analyzed in the Bio-Plex MAGPIX multiplex reader (Bio-Rad Laboratories Inc., CA). Fluorescence values were converted into cytokine concentrations (expressed as pg/mL), using a regression curve based on the values obtained from a set of standard dilutions.

The cytokines reported by the Bio-Plex Pro™ Human Cytokine panel were: Basic FGF, Eotaxin, G-CSF, GM-CSF, IFN-γ, IL-1β, IL-1Ra, IL-2, IL-4, IL-5, IL-6, IL-7, IL-8, IL-9, IL-10, IL-12, IL-13, IL-15, IL-17, IP-10, MCP-1, MIP-1α, MIP-1β, PDGF-BB, RANTES, TNF-α and VEGF.

## Statistical analysis and reproducibility

Each treatment was triplicated (PCR) or duplicated (all other experiments), and the experiments were repeated at least once to ensure reproducibility. Power analysis was done to determine the sample size to ensure biological significance. The data were analyzed using GraphPad Prism software. Unpaired student T-tests were used to analyze the differences between treated and control groups. All statistical data are represented as mean ± SEM. Statistical significance was defined as $*P \le 0.05$, $**P \le 0.01$, and $***P \le 0.001$. All source data used to generate figures has been deposited in Figshare (https://doi.org/10.6084/m9.figshare.25254571).

## Reporting summary

Further information on research design is available in the Nature Portfolio Reporting Summary linked to this article.

## Data availability

All data supporting the findings of this study are available in the manuscript (Supplementary Data 1). All source data in Figshare can be freely accessed (https://doi.org/10.6084/m9.figshare.25254571). If there are any special requests or questions for the data, please contact the corresponding author (G.Y.).

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

## Acknowledgements

We thank Dr. Amy Tvinnereim for helping perform the following experiments: *Mtb* infection of the humanized mice and the pulmonary function test. We wish to acknowledge support from the National Institute of Allergy and Infectious Diseases (UG3AI150550 to G.Y. and R21AI156798 to J.J.D.), and the National Heart, Lung, and Blood Institute (R01HL125016 to G.Y.).

## Author contributions

Guohua Yi: Conceived and guided the study, designed and performed experiments, analyzed data, edited figures, and wrote and finalized the manuscript. Fan Yang: Performed the experiments and analyzed data. Alireza Labani-Motlagh: Performed the experiments and analyzed data. Jose Alejandro Bohorquez Garzon: Performed the experiments and analyzed data. Josimar Dornelas Moreira: Performed the experiments. Danish Ansari: Performed the experiments. Sahil Patel: Performed the experiments. Fabrizio Spagnolo: Performed the experiments and edited the manuscript. Jon Florence: Performed the experiments. Abhinav Vankayalapati: Performed the experiments. Tsuyoshi Sakai: Performed the confocal miacroscopy experiments. Osamu Sato: Performed the confocal miacroscopy experiments. Mitsuo Ikebe: guided the confocal miacroscopy experiment. Ramakrishna Vankayalapati: Participated in experiment discussions, provided suggestions and edited the manuscript. John J. Dennehy: Designed experiments, participated in experiment discussions, provided phage strains, and edited the manuscript. Buka Samten: Designed experiments, guided experiments, performed experiments, participated in experiment discussions, and edited the manuscript.

## Competing interests

The authors declare no competing interests.
