## [Peer Review File · Communications Biology]

Reviewers' comments:

Reviewer #1 (Remarks to the Author):

In this work Fan Yang et al describe a phage treatment of Mtb in a humanized mouse model for tuberculosis. This model is important for replicating Tb granuloma formation. 1st Authors evaluate phage killing in a number of in vitro assays via top agar overlays and liquid cultures. 2nd Authors evaluate phage performance in primary macrophages. Finally, authors evaluate phage ability to reduce Mtb burden and TB symptoms in humanized mice, looking at a variety of metrics. Authors find a reduction or alleviation of symptoms in a number of areas, some rising to significance.

Findings are impactful and important for advancing phage therapy in the Mtb field; however, a lack of significance in a number of key metrics reduces its impact and claims to some extent, such as no significant reduction in lung associated Mtb. The development of mouse phage therapy model in which granuloma-like particles is perhaps most important finding, as it can be used for a variety of Mtb phage therapy tests. Some findings are perhaps novel in the specific lens of Mtb phage therapy but fairly generalizable to phage given via IV (such as high accumulation of phage in the spleen (and liver) and eventually generation of anti-phage Ab. Is there a reason phage not also given via aerosol/nebulizer to increase their access to the lungs? Perhaps this would have increased reduction in Mtb. In other mouse models, phage for K. pneumonia frequently applied intranasally/ via aerosol. There is some level of inconsistency in terms of controls as well, for some experiments it is PBS, for others a non-infecting phage had both been included in some claims could have been strengthened.

Specific Comments:

Line 28 – Claims three phages were tested for their Mtb killing activity, but in my mind really only two were “tested”. By authors own admission, Chah is known to not infect Mtb and was only included as a negative control. Consider revising to two phages D29 and DS6A were evaluated for therapeutic use.

Line 37 – “In 2020, while all focus was on COVID-19” This is a bit editorial and irrelevant? Implies some sort of causative effect? According, to WHO in 2016 Mtb infection and deaths globally were similar pre-COVID

Line 41 – “diseases/infections” prefer diseases or infections, then swap HIV and diabetes as they are out of order with which is the disease and which is the infection.

Line 43- increased – tense

Line 59 – “Phages” - Introduce term of Bacteriophage first, since it’s in the title before switching to shorthand phage. ie Bacteriophage, or phages, are

Line 68 – “lytic phage strains” - Perhaps personal preference, but “phage strains” cumbersome, simpler and more common to just say “lytic phages” or “lytic phage isolates”. Also, I presume by lytic you are meaning “virulent” phage? Temperate phage can also enter lytic cycle. ie all tailed phages are lytic

Line 70 – “reliable animal models” - Does this imply there have been unreliable animal models? If so, please cite.

Line 113 – “at one multiplicity of infection” – does this mean at a single MOI? Or at an MOI of 1? MM indicates it’s the latter, please clarify.

Line 122 – This line of experiments seems a bit redundant and less quantitative than previous assays (OD and CFU). Consider moving to supplemental.

Line 146 – Phage D29 was used as negative control, when previously phage Chah was the negative. When able negative should be kept consistent, unless needed. If the change in control was needed, please justify.

Line 146 – “Meanwhile” – This is a change in experimental design, meanwhile may not be appropriate as implies concurrent experiments. Consider using “Next” or other adjoining language

Line 165- are breeding – should be bred

Figure 4 – Please very explicitly clarify the number of mice (n) for each experiment, as I had difficulty finding this information in the text or MM. Further more in figure 4, there are differences in the number of data points on each graph, please indicate perhaps on each graph directly the n= is for each assay.

Line 184 – “and as confirmed by an average ~100 bacilli per lung deposited at day one post-infection” this is very wordy. It also does not confirm infection rather it is just stating the dosing concentration. Perhaps better to just state: “infected the humanized NSG-SGM3 mice with low dose aerosolized H37Rv of ~100 bacilli per lung”. Unless I miss understand and mice were actually sacrificed immediately after infection to determine number of bacilli that actually made it to the lung.

Line 185 – Here you state 10 doses in the MM you state 8, please clarify. ****

Line 209 - Phages given intravenously will often end up being filtered out and concentrated into the spleen and liver. Perhaps a citation to this effect, there should be many available.

Not sure claims can be made regarding replication in the spleen, this could simply be due the high dosing and concentrating of phage into the spleen. This is where a nonreplicating control phage would have been beneficial such as Chah, then stronger claims on replication of phage could be made. Or a phage only control, to compare the actually level of phage replication.

Line 153 – Phage killing Mtb in macrophages. There is a possibility this killing effect occurred outside of the macrophages as the samples were homogenized and phage bound their host at this point when plated. Especially if the macrophages were not sufficiently washed prior to cell lysis to remove residual phage and phage were added at high MOI. Please clarify in the MM, as it was unclear to me if residual phage was removed or consider adding a line indicating this possibility as well.

To this end, an additional experiment rule this out would be to repeat it with the H37Rv-GFP and look for

GFP signal within the macrophages. If GFP signal depreciates, phage truly are in killing Mtb within the macrophages.

Line 253 - Reword, "increased effectiveness is too broad", what you are referring to sustained phage levels due to replication, but that does not necessarily mean increased effectiveness compared to antibiotics.

Line 271 – "shown capable" – shown to be capable of

Line 279 – Phage D29 and DS6A both appear to be temperate, as are most identified mycobacterium phages. A quick scan of their genbank records show each has an integrase. Is it known if there are capable of lysogenizing H37Rv? An alternative interpretation could be that D29, if it has a high rate of integration forms lysogens which are insensitive to superinfection by the same phage. If this occurs early enough, its growth curve would appear like that of the bacteria only control. Authors should confirm if these phages are capable of lysogen formation in H37Rv, it may provide insight into a number of the observed phenotypes. This can be achieved a number of ways, PCR on surviving colonies, or screen for spontaneous induction of phage particles from surviving H37Rv after overnight growth (or over week in the case of Mtb)

Please also address their temperate nature somewhere in the text as it is highly relevant to therapeutic phages, classically only virulent phages should be used. DS6A may require engineering to be a plausible phage for therapy as Graham Hatfull's group has been attempting removing genes for lysogeny.

Line 302 – Again, please elaborate on the decision to introduce phage via IV rather than aerosol/nebulization.

Line 310 – "and was undetectable after 36 h of injection" Please fix, should read something like: Some studies have reported administered phage were undetectable after 36hrs.

Line 341-345 – please revise sentence structure – its lists questions remaining to be asked except last is a statement vs a ?

Line 371 – " A hundred mLs into 5 mL"? This is presumable meant to be ul

Line 380 – In our experience freezing of phages (especially without cryoprotectant) can be deleterious to titer

Line 456 – Again please specify more explicitly the number of mice, and n for each of the different assays.

Line 487/Figure 5 – If serum still available, data on if the IgM is neutralizing to phage infection may be relevant to Ig inhibition of phage therapy. Given the prolonged nature of this model, sera taken at regular intervals to track phage specific Ab emergence would be informative. Furthermore, if sera still available, testing sera against another mycobacterium phage (perhaps D29) could serve as an interesting

control for cross reactivity and specificity of IgM.

Fig 2B – These seem to be rather modest differences in CFU with the exception of DS6A, generally working phage produce reductions on the Log scale, I could imagine these be explained from counting error depending on the dilution, number of replicates, and colonies counted.

While I feel the few suggested follow up experiments/controls could strengthen claims, they are at the authors' discretion as they may be impractical given animals are involved and slow growing MtB, but the additional details and requested clarifications should be addressed.

Reviewer #2 (Remarks to the Author):

Tuberculosis is indeed a global disease associated with significant mortality. The challenges to treat tuberculosis are well documented and alternative therapeutic approaches to successfully intervene is critically needed. The manuscript submitted by Yan et al, investigates M. tuberculosis killing using bacteriophages as a potential alternative therapy to antibiotic use in view of the emergence of drug resistance. Studies were conducted in culture and employed a humanized mouse model to investigate phage therapy in vivo. The study claims treatment efficacy with the phage strain DS6A, demonstrated by reduced bacilli burdens in mouse organs, bodyweight gain and improved lung function.

MAJOR COMMENTS

1. One of the primary outcomes to demonstrate phage therapy as a potential therapeutic approach to treat tuberculosis is to demonstrate that it can target the lungs and reduce bacilli burden. This study failed to demonstrate significant reduction in pulmonary bacilli burden. The data does not contradict the null hypothesis and therefore does not support differences for the lung in this study. The study design may well have influenced the outcome e.g. the duration of treatment but as it stands the authors cannot claim differences between the pulmonary burdens for treated and untreated conditions. The intravenous administration of phages under the experimental conditions of this study favours targeting the spleen over the lungs, and the authors demonstrate this through the higher copy numbers present in the spleen. Thus, it appears that the route of administration may be a significant factor that inhibit phages to reach the lungs and limiting its efficacy to kill pulmonary bacilli. In fact, the authors acknowledge the limitation of the study in the Discussion and proposes optimisation of administration. Thus there appear to be an over-reach in the conclusions made based on the data from this study (e.g “Phage DS6A effectively eradicates Mtb in humanized mice” Line 183 – heading)
2. Due to the equivalence of pulmonary bacilli burdens, one would have expected similar degrees of inflammation to be present in treated and untreated mice. How do the authors account for the improved lung function? Does phage treatment induce effects other than bacilli killing?
3. (Fig 2B: The Mtb CFU values at d10 (red bar) are almost 3 fold less than that of D29 and Chah. This is unusual and suggest that the presence of D29 and Chah stimulates mycobacterial growth. The argument put forward by the authors in the Discussion (Line 289-295) is not convincing, nor do the authors provide

reference to previous such findings.

4. (Fig 4C) Did the authors consider a longer phage treatment period? On what basis were 20 days selected as the treatment period?

5. (Fig 4E). Quantification of density and images of more than 1 mouse would strengthen the data.

6. The use of humanised mice in this study is commendable.

GENERAL COMMENTS

Page 4, Line 80: The authors should comment on findings on pulmonary pathology observed Kramnik mice.

The quality of image 1e, 2c 2e should be improved. Penlines in images submitted for publication should be removed.

Reviewer #3 (Remarks to the Author):

In this manuscript, Yang & Labani-Motlagh et. al. tested the infectivity of three phages in Mtb cells via plate assays, liquid culture, and Mtb-infected macrophages and established phage DS6A as a candidate for phage therapy. The authors then infected humanized mice NSG-SGM3 with Mtb, which mimics Mtb infection in human, and treated the infected mice with phage DS6A. Authors observed that DS6A efficiently eradicated Mtb cells in mice spleen while had a minor killing effect of Mtb cells in mice lung. Nonetheless, mice that received DS6A treatment enjoyed better health conditions than those that did not receive the phage treatment, supporting the possibility of applying phage therapy in TB patients. Overall, the authors dissected the problem systematically and the manuscript is well written.

One major caveat is the narrow range of MOI when it comes to phage infection that was applied in the current study. Since phage infection is often MOI-dependent, it is advised to test several magnitudes of phage MOI (i.e. MOI 0.01 to MOI100 at 10-fold intervals) before reaching the qualitative conclusion of whether a phage infects a specific bacterial species or not. While it could be costly and time-consuming to test different MOIs in mice models, authors should consider broadening the MOI range in in vitro assays. Below are detailed comments:

Line 105: It is unclear why these three phages were chosen. What is the origin of these three phages? How many phages have been identified in killing Mtb? What is the rationale for testing these three phages?

Line 107: What is the MOI?

Line 117: MOI is important in killing bacteria. Even though D29 at MOI 1 does not kill Mtb in liquid culture. D29 at higher MOIs might. It is worth covering a broader range of MOIs when testing phage infectivity.

Figure 2: The figure legend was very helpful for one to understand the experimental procedures and parameters. However, most of the legend details were not included in the main text. Authors should

consider repeating/rephrasing figure legends in the main text so the audience does not have to dig into the figure legend to find experimental details.

Figure 3(A): What is the difference between the top and bottom panels? For instance, why are there two panels for SSCH X FSCH?

Figure 4(E): Quantify and compare the high-density areas without and with phage treatment. I am not familiar with CT scan images. To me, it is not obvious that these two images are different.

Line 275-277: The infectivity of D29 was not tested in humanized mice to support the authors' claim.

Line 281: Are Mtb cells used in liquid culture non-clonal? If true, could authors repeat the experiments with clonal Mtb cells?

Responses to Reviewers' Comments

First of all, we really appreciate the reviewers for raising constructive critiques and comments of our manuscript, which are very helpful for us to improve the manuscript. Now we address the reviewers' comments point-by-point, and we also reflected them in the revised manuscript.

Response to Reviewer #1

Line 28 – Claims three phages were tested for their Mtb killing activity, but in my mind really only two were “tested”. By authors own admission, Chah is known to not infect Mtb and was only included as a negative control. Consider revising to two phages D29 and DS6A were evaluated for therapeutic use.

We thank the reviewer for the excellent comment. We revised to two phages D29 and DS6A.

Line 37 – “In 2020, while all focus was on COVID-19” This is a bit editorial and irrelevant? Implies some sort of causative effect? According, to WHO in 2016 Mtb infection and deaths globally were similar pre-COVID

The line was intended to show the impact that the COVID-19 pandemic had in other infectious diseases, as directly established by the 2022 WHO Global TB report: “The coronavirus (COVID-19) pandemic has caused enormous health, social and economic impacts since 2020. This includes impacts on the provision of and access to essential tuberculosis (TB) services, the number of people diagnosed with TB and notified as TB cases through national disease surveillance systems, and TB disease burden (incidence and mortality)”

However, after careful consideration, we feel that it may cause confusion, thus we delete “In 2020, while all focus was on COVID-19”.

Line 41 – “diseases/infections” prefer diseases or infections, then swap HIV and diabetes as they are out of order with which is the disease and which is the infection.

This has been fixed.

Line 43- increased – tense

This has been fixed.

Line 59 – “Phages” - Introduce term of Bacteriophage first, since it’s in the title before switching to shorthand phage. ie Bacteriophage, or phages, are ...

This has been fixed.

Line 68 – “lytic phage strains” - Perhaps personal preference, but “phage strains” cumbersome, simpler and more common to just say “lytic phages” or “lytic phage isolates”. Also, I presume by lytic you are meaning “virulent” phage? Temperate phage can also enter lytic cycle. ie all tailed phages are lytic

This has been fixed.

Line 70 – “reliable animal models” - Does this imply there have been unreliable animal models? If so, please cite.

The statement refers to the difficulty in obtaining a humanized mouse model that can recapitulate the diversity and function of the immune cell phenotypes found in humans. We compared the commonly used NSG background mice with the NSG-SGM3 background mice, and extensively demonstrated the advantages of NSG-SGM3 mice used in our study vs NSG background mice with substantial citations.

Considering the reviewer’s comment, we changed wording to “appropriate”.

Line 113 – “at one multiplicity of infection” – does this mean at a single MOI? Or at an MOI of 1? MM indicates it’s the latter, please clarify.

This has been modified to clarify.

Line 122 – This line of experiments seems a bit redundant and less quantitative than previous assays (OD and CFU). Consider moving to supplemental.

We appreciate the reviewer’s comment. However, we believe that this assay provides additional validation for the selection of DS6A as our phage therapy candidate and, as such, fits within the scope of the paper. We would prefer to keep this experiment in the manuscript.

Line 146 – Phage D29 was used as negative control, when previously phage Chah was the negative. When able negative should be kept consistent, unless needed. If the change in control was needed, please justify.

We agree with the reviewer that the negative control should be consistent. We used both D29 and Chah as negative control in the first donor of macrophage intracellular Mtb-killing assay (Fig 2b and 2c). After we found that the phage DS6A cannot kill Mtb in macrophages, then we used only phage D29 as a stricter negative control in the

following experiments. Our justification was to reduce the resources and labor required to do the experiments. However, in the Fig. 4f experiment that was used to test the phage DS6A resistance, we used Chah again as negative control, because phage D29 can kill Mtb on agar plate, and therefore it cannot serve as a negative control.

Line 146 – “Meanwhile” – This is a change in experimental design, meanwhile may not be appropriate as implies concurrent experiments. Consider using “Next” or other adjoining language

This has been fixed.

Line 165- are breeding – should be bred

This has been fixed.

Figure 4 – Please very explicitly clarify the number of mice (n) for each experiment, as I had difficulty finding this information in the text or MM. Further more in figure 4, there are differences in the number of data points on each graph, please indicate perhaps on each graph directly the n= is for each assay.

We added the number for each experiment in Figure 4 legend.

Line 184 – “and as confirmed by an average ~100 bacilli per lung deposited at day one post-infection” this is very wordy. It also does not confirm infection rather it is just stating the dosing concentration. Perhaps better to just state: “infected the humanized NSG-SGM3 mice with low dose aerosolized H37Rv of ~100 bacilli per lung”. Unless I miss understand and mice were actually sacrificed immediately after infection to determine number of bacilli that actually made it to the lung.

The text has been modified to clarify in the results section and in the materials and methods section. Three additional animals were included in the inoculation and were euthanized on the following day. Lungs were collected and macerated, and serial dilutions of the macerate were plated in 7H10 agar to determine the Mtb CFU load. This is how the bacilli per lung number was obtained.

*Line 185 – Here you state 10 doses in the MM you state 8, please clarify. *****

Thanks to the reviewer for bringing to our attention this mistake. It was 10 doses of treatment. It has been corrected in the materials and methods section.

Line 209 - Phages given intravenously will often end up being filtered out and concentrated into the spleen and liver. Perhaps a citation to this effect, there should be

many available. Not sure claims can be made regarding replication in the spleen, this could simply be due the high dosing and concentrating of phage into the spleen. This is where a nonreplicating control phage would have been beneficial such as Chah, then stronger claims on replication of phage could be made. Or a phage only control, to compare the actually level of phage replication.

We agree with the reviewer. We added citation to address this effect. However, due to the shortage of the humanized mice and getting a new batch of mice needs >6 months, we were not able to address this phage replication issue. However, it is safe to claim the distribution. So, we reworded the sentence, and we believe now, our claim of different distribution is appropriate.

Line 153 – Phage killing Mtb in macrophages. There is a possibility this killing effect occurred outside of the macrophages as the samples were homogenized and phage bound their host at this point when plated. Especially if the macrophages were not sufficiently washed prior to cell lysis to remove residual phage and phage were added at high MOI. Please clarify in the MM, as it was unclear to me if residual phage was removed or consider adding a line indicating this possibility as well.

To this end, an additional experiment rule this out would be to repeat it with the H37Rv-GFP and look for GFP signal within the macrophages. If GFP signal depreciates, phage truly are in killing Mtb within the macrophages.

We agree with the reviewer. **We supplemented an experiment** using H37Rv-GFP as the reviewer suggested and the **revised figure 2f** showed our new confocal microscopy results, which further confirmed the intracellular Mtb-killing of DS6A. We added the results and methods in the revised manuscript as well.

Line 253 - Reword, “increased effectiveness is too broad”, what you are referring to sustained phage levels due to replication, but that does not necessarily mean increased effectiveness compared to antibiotics.

This has been fixed.

Line 271 – “shown capable” – shown to be capable of

This has been fixed.

Line 279 – Phage D29 and DS6A both appear to be temperate, as are most identified mycobacterium phages. A quick scan of their genbank records show each has an integrase. Is it known if there are capable of lysogenizing H37Rv? An alternative

interpretation could be that D29, if it has a high rate of integration forms lysogens which are insensitive to superinfection by the same phage. If this occurs early enough, its growth curve would appear like that of the bacteria only control. Authors should confirm if these phages are capable of lysogen formation in H37Rv, it may provide insight into a number of the observed phenotypes. This can be achieved a number of ways, PCR on surviving colonies, or screen for spontaneous induction of phage particles from surviving H37Rv after overnight growth (or over week in the case of Mtb)

Please also address their temperate nature somewhere in the text as it is highly relevant to therapeutic phages, classically only virulent phages should be used. DS6A may require engineering to be a plausible phage for therapy as Graham Hatfull's group has been attempting removing genes for lysogeny.

While D29 does have an intact attP-integration system, it has lost the repressor gene required for maintenance of lysogeny and superinfection immunity. Hence we feel the most likely explanation for our results is that Mtb rapidly acquired resistance to D29 and the resistant cells dominated the liquid culture.

Although DS6A does contain genes associated with lysogeny, but lysogeny has not been experimentally demonstrated. Moreover, DS6A does not form turbid plaques on host lawns suggesting that if it is capable of lysogeny, it does so at a low rate.

The reviewer's point is well taken, and we have revised the manuscript to include a more detailed discussion of the potential for lysogeny in these phage strains. Moreover, we bring up the reviewer's good point that the lysogeny related genes should be engineered out of the phage genome in future work. See lines 507-533.

Line 302 – Again, please elaborate on the decision to introduce phage via IV rather than aerosol/nebulization.

Our original intent was to deliver phage via a nebulizer and we went as far as constructing a nebulizing chamber to dose the mice. However, our initial experiments with this chamber were not promising, and we reverted to IV delivery until we can improve our ability to dose the mice via aerosol/nebulization delivery.

At any rate, even though the target organ for Mtb is the lung, the infection can spread to different organs, as evidenced in our results, as well as the literature. So we believe that a systematic delivery will be needed to eradicate the bacilli throughout the body, and it is likely that a combination of aerosol and IV dosing will be necessary for the complete eradication of Mtb. We have ongoing research projects to develop aerosol/nebulization delivery and the results will be published separately.

Line 310 – “and was undetectable after 36 h of injection” Please fix, should read

something like: Some studies have reported administered phage were undetectable after 36hrs.

The phrase has been modified to clarify the results.

Line 341-345 – please revise sentence structure – its lists questions remaining to be asked except last is a statement vs a ?

The phrase has been modified to clarify the results.

Line 371 – “ A hundred mLs into 5 mL”? This is presumable meant to be ul

This has been fixed.

Line 380 – In our experience freezing of phages (especially without cryoprotectant) can be deleterious to titer

We appreciate the reviewer's comment. We have been keeping the phages at 4°C for short to medium term storage.

Line 456 – Again please specify more explicitly the number of mice, and n for each of the different assays.

We specified the number of mice in the Figure 2 legend.

Line 487/Figure 5 – If serum still available, data on if the IgM is neutralizing to phage infection may be relevant to Ig inhibition of phage therapy. Given the prolonged nature of this model, sera taken at regular intervals to track phage specific Ab emergence would be informative. Furthermore, if sera still available, testing sera against another mycobacterium phage (perhaps D29) could serve as an interesting control for cross reactivity and specificity of IgM.

We thank the reviewer for their suggestion. We have some DS6A-treated mouse sera left, and then we **supplemented an experiment to test the neutralization** of phage DS6A using these sera. The result is shown in **Supplementary figure 4**. We also added the methods of neutralization assay and added several sentences in the Results section.

Fig 2B – These seem to be rather modest differences in CFU with the exception of DS6A, generally working phage produce reductions on the Log scale, I could imagine these be explained from counting error depending on the dilution, number of replicates, and colonies counted.

While I feel the few suggested follow up experiments/controls could strengthen claims, they are at the authors' discretion as they may be impractical given animals are involved and slow growing MtB, but the additional details and requested clarifications should be addressed.

We agree with the reviewer, so we confirmed the CFU counting in Fig 2B, and our claim that on Day 10, the MtB bacilli can be completely eliminated by DS6A should be appropriate. This conclusion was further confirmed in 4 donors' PBMCs.

Response to Reviewer #2

1. One of the primary outcomes to demonstrate phage therapy as a potential therapeutic approach to treat tuberculosis is to demonstrate that it can target the lungs and reduce bacilli burden. This study failed to demonstrate significant reduction in pulmonary bacilli burden. The data does not contradict the null hypothesis and therefore does not support differences for the lung in this study. The study design may well have influenced the outcome e.g. the duration of treatment but as it stands the authors cannot claim differences between the pulmonary burdens for treated and untreated conditions. The intravenous administration of phages under the experimental conditions of this study favours targeting the spleen over the lungs, and the authors demonstrate this through the higher copy numbers present in the spleen. Thus, it appears that the route of administration may be a significant factor that inhibit phages to reach the lungs and limiting its efficacy to kill pulmonary bacilli. In fact, the authors acknowledge the limitation of the study in the Discussion and proposes optimisation of administration. Thus there appear to be an over-reach in the conclusions made based on the data from this study (e.g "Phage DS6A effectively eradicates MtB in humanized mice" Line 183 – heading)

We totally agree with the reviewer for this excellent comment. We modified our claim to reflect the impact of intravenous administration of phage DS6A in the previous Line 183 – heading.

2. Due to the equivalence of pulmonary bacilli burdens, one would have expected similar degrees of inflammation to be present in treated and untreated mice. How do the authors account for the improved lung function? Does phage treatment induce effects other than bacilli killing?

Thanks to the reviewer and this is a very good point. Although the lung bacilli burden between the MtB control mice and the phage-treated mice was not significant, the P value (0.0565) is very close to significance. Given that we showed the bacilli burden in 10 log scale, we reason that the less bacilli burden in phage-treated group (~8-fold less

than in Mtb control group) contributed to the improved lung function. However, we cannot exclude any other reason for the improved lung function. We added several sentences in the discussion section.

3. (Fig 2B: The Mtb CFU values at d10 (red bar) are almost 3 fold less than that of D29 and Chah. This is unusual and suggest that the presence of D29 and Chah stimulates mycobacterial growth. The argument put forward by the authors in the Discussion (Line 289-295) is not convincing, nor do the authors provide reference to previous such findings.

Now we provided another explanation with a citation.

4. (Fig 4C) Did the authors consider a longer phage treatment period? On what basis were 20 days selected as the treatment period?

We believe that the optimization of the regimen is definitely an excellent suggestion, and we will try it in the future. In this paper, we performed a pilot experiment and found that 20 days is enough to eradicate the splenic bacilli, so we set 20 days as our treatment period.

5. (Fig 4E). Quantification of density and images of more than 1 mouse would strengthen the data.

We agree with the reviewer, and now we added one more mouse lung image for each group and reflected it in **Revised Fig. 4**.

6. The use of humanised mice in this study is commendable.

Thanks to the reviewer for the praise and encouragement.

GENERAL COMMENTS

Page 4, Line 80: The authors should comment on findings on pulmonary pathology observed Kramnik mice.

Thanks to the reviewer for the important reminder. We commented on the Kramnik mice for the capability of mimicking the Mtb-caused pulmonary pathology in humans.

The quality of image 1e, 2c 2e should be improved. Penlines in images submitted for publication should be removed.

We changed the representative figure in Fig. 2e with significant Red penline. However, all the penlines were used to mark the bacilli dots, it is difficult to remove them all. We

believe the revised figure looks better and I hope the reviewer can forgive this minor issue.

Response to Reviewer #3

Line 105: It is unclear why these three phages were chosen. What is the origin of these three phages? How many phages have been identified in killing Mtb? What is the rationale for testing these three phages?

We chose D29 and DS6A because we know they are effective for Mtb-killing according to the literature. The Chah was chosen as negative control. The original sources were described in the Bacterial strains and phage strains of the Methods sections

Line 107: What is the MOI?

The MOI had been indicated in the materials and methods section, but has been added to the results section in accordance with the reviewer's concern

Line 117: MOI is important in killing bacteria. Even though D29 at MOI 1 does not kill Mtb in liquid culture. D29 at higher MOIs might. It is worth covering a broader range of MOIs when testing phage infectivity.

We agree with the reviewer that we cannot exclude this possibility. Thus, we supplemented an experiment to test if this possibility. Our result showed that even using a high MOI of D29 (MOI of 100) still could not Mtb bacilli. The result is shown in **Supplementary Figure 1**.

Figure 2: The figure legend was very helpful for one to understand the experimental procedures and parameters. However, most of the legend details were not included in the main text. Authors should consider repeating/rephrasing figure legends in the main text so the audience does not have to dig into the figure legend to find experimental details.

We agree with the reviewer regarding the suggestion and added the key parameters in the text.

Figure 3(A): What is the difference between the top and bottom panels? For instance, why are there two panels for SSCH X FSCH?

We thank the reviewer for spotting this inconsistency. The top panels correspond with splenocytes and the lower panels with PBMC. We have included this distinction in the figure legend.

Figure 4(E): Quantify and compare the high-density areas without and with phage treatment. I am not familiar with CT scan images. To me, it is not obvious that these two images are different.

The image analysis was performed by CT scan expert in our campus, and we added another mouse representative figure which show more clear difference.

Line 275-277: The infectivity of D29 was not tested in humanized mice to support the authors' claim.

The paragraph has been rephrased to comply with the reviewer's comment.

Line 281: Are Mtb cells used in liquid culture non-clonal? If true, could authors repeat the experiments with clonal Mtb cells?

The Mtb cells used are clonal, as detailed in reference: 10.1101/gr 166401 Genome Res. 2001. 11: 547-554

Reviewers' comments:

Reviewer #1 (Remarks to the Author):

In general, authors have satisfactorily responded to my comments and addressed them and/or justified their reasoning, even so far as to doing additional experiments which strengthen their claims. However, a few caveats remain.

While phage killing inside macrophages is much more convincing now, a few lingering comments regarding the experimental design of Figure 2F, and will leave to the authors expertise regarding Mtb. Is the 4 hr treatment long enough for Mtb to efficiently infiltrate the macrophages? If not, is it possible Mtb cells that remain after washing go on to infiltrate the macrophages during the next 7 day incubation period. Washing rarely removes ALL bacterial cells, even a few Mtb remaining after wash could go on to infiltrate and propagate within the cells. If this is the case, phage treatment of these extracellular Mtb cells which remain after washing are still killed in their extracellular state when phage are added ~4hrs later. This could similarly lead to no GFP in phage treatment and GFP signal in the no phage control. A perhaps even more convincing design would have been to visualize the developing GFP signal within macrophages in all treatment groups, then add phage (or no phage), and observe the GFP signal dissipate. This would definitely show and confirm that Mtb was intracellular at the time it was killed by phage infection.

Furthermore, regarding the possible temperate nature of the two phages: If D29's repressor has not been specifically engineered out, or there is a nearly identical phage with a repressor displaying an obvious deletion, it is possible the repressor is merely cryptic and not recognizable. This does not necessarily indicate no integration or lysogeny. We have observed cases in which stable lysogens are formed but no repressor identifiable. Also. rebuttal document claims text regarding lysogeny has been added at "See lines 507-533"; however, I see no such discussion in the new text at this position (nor does a search of "temperate", "lysogeny", "integrase", "repressor" yield hits to any new text). Perhaps this was a saving/uploaded older document version mistakenly?

Reviewer #2 (Remarks to the Author):

All comments have been addressed.

Responses to Reviewers' Comments

We appreciate the reviewers for the favorable review and raising constructive critiques of our revision, which are very helpful for us to further improve the manuscript. Now we address the reviewers' comments point-by-point, and we also reflected them in the revised manuscript.

Response to Reviewer #1

In general, authors have satisfactorily responded to my comments and addressed them and/or justified their reasoning, even so far as to doing additional experiments which strengthen their claims. However, a few caveats remain.

1. While phage killing inside macrophages is much more convincing now, a few lingering comments regarding the experimental design of Figure 2F, and will leave to the authors expertise regarding Mtb. Is the 4 hr treatment long enough for Mtb to efficiently infiltrate the macrophages? If not, is it possible Mtb cells that remain after washing go on to infiltrate the macrophages during the next 7 day incubation period. Washing rarely removes ALL bacterial cells, even a few Mtb remaining after wash could go on to infiltrate and propagate within the cells. If this is the case, phage treatment of these extracellular Mtb cells which remain after washing are still killed in their extracellular state when phage are added ~4hrs later. This could similarly lead to no GFP in phage treatment and GFP signal in the no phage control. A perhaps even more convincing design would have been to visualize the developing GFP signal within macrophages in all treatment groups, then add phage (or no phage), and observe the GFP signal dissipate. This would definitely show and confirm that Mtb was intracellular at the time it was killed by phage infection.

We thank the reviewer for the excellent comments and reasonable concern. When we supplemented the experiment, we have performed experiment to ensure the intracellular Mtb infection in macrophage after 4 h-post infection (but this data have not presented last time, since we thought the confocal data is more beautiful and is sufficient to support our claim). Now we present the fluorescence microscopy data here (NOT confocal microscopy as in Fig 2f). We can clearly localize the intracellular Mtb after 4 h of Mtb infection followed by 3 times of rigorous washing (white arrows showing the intracellular Mtb), while no extra bacteria found in the microscopy field (See **New Supplementary Fig. 3** in the manuscript, and shown below). We added the following sentence in the text “We confirmed that *Mtb* bacilli entered the macrophages at 4h post-infection using fluorescence microscopy (**Supplementary Fig. 3**).” (Line 168-170)

In addition, we have extensively characterized the time needed for *Mtb* to enter human macrophages. In another experiment, we infected the macrophages with different MOI of *Mtb*, and at different time points, we used acid-fast stain (a classical experiment to visualize *Mtb* for clinical diagnosis) to visualize *Mtb*, we found that THREE hours is enough for *Mtb* entering the macrophages (see the following figure).

Peripheral blood monocyte derived macrophages (MDMs) from healthy donors were incubated with *Mycobacterium tuberculosis* H37Rv-GFP (*MtbH37Rv*) at different MOIs for four hours and the cells were washed with Hanks balanced salt solution (HBSS) to remove extracellular bacilli. The cells were then fixed with 4% paraformaldehyde in PBS over night. The cells were then washed with HBSS and stained with acid fast stain and the images of the cells were taken under a microscope at 1,000X magnification for the visualization of *Mtb H37Rv*. The cells positive for *Mtb* were determined by randomly counting 50 to 100 cells including both *Mtb* positive and negative cells for the determination of MDM percent phagocytosis of acid-fast positive bacilli. MDMs without (A), with *Mtb H37Rv* (B) and the mean \pm SEM of experiments with MDMs from five different donors (C) are shown.

2. Regarding the possible temperate nature of the two phages: If D29's repressor has not been specifically engineered out, or there is a nearly identical phage with a repressor displaying an obvious deletion, it is possible the repressor is merely cryptic and not recognizable. This does not necessarily indicate no integration or lysogeny. We have observed cases in which stable lysogens are formed but no repressor identifiable. Also, rebuttal document claims text regarding lysogeny has been added at "See lines 507-533"; however, I see no such discussion in the new text at this position (nor does a search of "temperate", "lysogeny", "integrase", "repressor" yield hits to any new text). Perhaps this was a saving/uploaded older document version mistakenly?

We thank the reviewer for the excellent comment. We add the following sentences to address the reviewer's concern: Alternatively, it is possible that D29 is capable of lysogenizing *Mtb*, but the available evidence suggests that this is not the case. First, D29 does not form turbid plaques on host lawns suggesting that, if it is capable of lysogeny, it does so at a very low rate (Fig. 1). Moreover, although D29 does have an intact attP-integration system, no repressor gene required for the maintenance of lysogeny and superinfection has been identified. While it is possible that D29 possesses a cryptic or unrecognizable repressor, a genome analysis suggests that it does not. D29 is highly homologous to the mycobacteriophage L5, but has a 3.6 kb deletion relative to L5 that removes a part of its genome that corresponds to L5's repressor gene (Also added in the text, Line 314-322).

REVIEWERS' COMMENTS:

Reviewer #1 (Remarks to the Author):

The authors have sufficiently satisfied my few lingering concerns. I believe the additional supplemental material and additional genome analysis strengthens their claims.